# RECONCILING VISUAL PERCEPTION AND GENERATION IN DIFFUSION MODELS

**Liulei Li[1], Yi Yang[1], Wenguan Wang[1†]**
[1]The State Key Lab of Brain-Machine Intelligence, Zhejiang University

## ABSTRACT

We present GENREP, a unified image understanding and synthesis model that jointly conducts discriminative learning and generative modeling in one training session. By leveraging Monte Carlo approximation, GENREP distills distributional knowledge embedded in diffusion models to guide the discriminative learning for visual perception tasks. Simultaneously, a semantic-driven image generation process is established, where high-level semantics learned from perception tasks can be used to inform image synthesis, creating a positive feedback loop for mutual boosts. Moreover, to reconcile the learning process for both tasks, a gradient alignment strategy is proposed to symmetrically modify the optimization directions of perception and generation losses. These designs empower GENREP to be a versatile and powerful model that achieves top-leading performance on both image understanding and generation benchmarks. Our code is available at GENREP.

## 1 INTRODUCTION

Broadly, there are two fundamental goals in the field of computer vision: visual understanding which extracts meaningful cues from scenes, and image generation which aims to create new visual contents. The former is typically solved through visual representation learning, *i.e.*, transforming raw pixel data into features or embeddings that can capture high-level semantics (Bengio et al., 2013) in a discriminative manner. This leads to strong performance in downstream tasks such as visual recognition and semantic segmentation. On the other hand, image generation relies on generative modeling and emphasizes the learning of underlying patterns and distributions within data (Croitoru et al., 2023), thereby enabling the synthesis of new samples that faithfully resemble the original one.

Since visual understanding and synthesis have long been addressed with different paradigms, most existing work excels in either synthesizing realistic outputs or interpreting input data, but seldom do both on a unified basis. This brings several drawbacks: ① Representations learned in a discriminative manner for visual perception tasks often generalize poorly to unseen patterns (Pourpanah et al., 2022) and overlook fine-grained details (Huynh & Elhamifar, 2020). This stems from their narrow focus on decision boundary between classes (Jebara, 2012), rather than capturing the underlying data distribution like generative models. ② Modern generative models such as GANs (Goodfellow et al., 2014) or diffusion models (Sohl-Dickstein et al.; Rombach et al., 2022) exhibit a lower-level understanding of semantics due to the reliance on low-level reconstruction loss (Zhang et al., 2023a). As a result, they tend to underperform discriminative approaches in scene understanding tasks. ③ The divergence in technological protocols for image understanding and synthesis diffuses the research endeavors, and hinders innovations and insights achieved in one paradigm to enhance the other.

This stimulates us to rethink the perceived incompleteness in discriminate-based representation learning and generative modeling, and seek to bridge this gap by preserving both synthesis and understanding abilities within the same model. Our idea is motivated by the observations that: **i)** diffusion models facilitate downstream visual perception tasks (Zhao et al., 2023; Yang & Wang, 2023); **ii)** high-quality discriminative representations accelerate the generative learning of diffusion models (Yu et al., 2024). This reveals the potential commonality of representations learned via two paradigms, and forms the basis for devising a unified visual understanding and generation framework.

---

† Corresponding Author: Wenguan Wang.

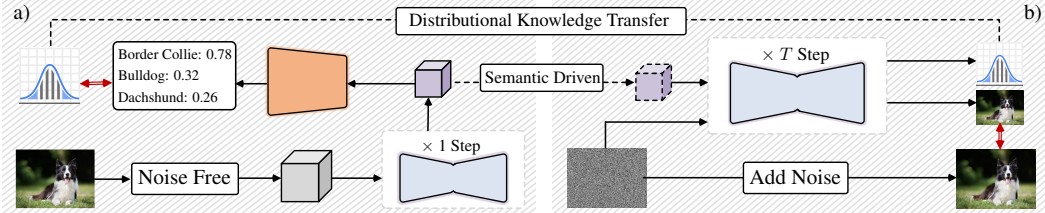

Figure 1: Unified image Understanding and synthesis within diffusion models: a) semantic driven image generation; b) distributional knowledge transfer from diffusion models for visual perception.

Building on this premise, we introduce GENREP, which reconciles the learning processes of downstream visual perception tasks and image generation in diffusion models while enabling the mutual benefits. **First**, to enhance visual understanding, GENREP leverages the distributional knowledge captured by generative modeling (Fig. 1(a)). Assuming the diffusion-based image generation can capture object distributions $p(\boldsymbol{x}|y)$ with class labels as conditional inputs, we approximate it through Markov Chain Monte Carlo where intermediate outputs during reverse diffusion are utilized as representative samples. As such, the class-wise posterior probabilities $p(y|\boldsymbol{x})$ can be retrieved via Bayes' theorem, which serves as a supplementary guidance for the discriminative learning of downstream perception tasks. **Second**, to enable image generation informed by visual understanding, we propose a semantic-driven generation learning strategy to guide image synthesis with high-level semantics derived from perception tasks (Fig. 1(b)). It refines the reverse diffusion process by conditioning the noise distribution on semantic embeddings delivered by the perception branch, encouraging the generated images to faithfully reflect the desired content. **Finally**, GENREP fosters the iteratively mutual enhancement between visual perception and image generation through a joint optimization strategy, which aligns the gradient of the generation loss with the direction of the perception loss at each training step. This aims to harmonize the learned representations for both tasks, so as to deliver a single and cohesive model capable of effectively tackling both visual perception and generation.

By exploring the interplay between visual perception and generation, GENREP offers several compelling advantages over disjointed paradigms: **First**, unlike prior diffusion-based work (Zhu et al., 2024a; Xu et al., 2023; 2024) that often compromises the generation ability for visual understanding, our approach holds superior performance for both tasks. **Second**, it moves from purely deterministic modeling to joint discriminative and generative learning, thereby demonstrating notably low expected calibration errors and benefits perception under open-vocabulary scenarios. **Third**, through joint optimization and gradient alignment, a feedback loop is established, where the unique strengths of two learning paradigms can be leveraged to enhance each other. **Fourth**, the construction of a shared feature space for both perception and generation tasks facilitates the emergence of more robust and transferable representations, which improves the generalization across a variety of downstream tasks.

For thorough examination, we experiment GENREP on both visual perception and generation tasks. It consistently demonstrates remarkable performance across benchmarks, including **57.8** for out-of-the-distribution generalization on ObjectNet (Barbu et al., 2019), **92.9** for fine-grained classification on CUB-200 (Wah et al., 2011), **0.057** AbsRel for monocular depth estimation on NYUv2 (Silberman et al., 2012), **34.7/54.6** mIoU for open/close-set semantic segmentation on ADE-20K (Zhou et al., 2017), and **56.5/36.0** AP for open-vocabulary object detection on MS COCO (Lin et al., 2014)/LViS v1.0 (Gupta et al., 2019), in leverage of advanced diffusion architectures such as CNN-based Latent Diffusion Models (LDM) (Rombach et al., 2022) and ViT-based Diffusion Transformers (DiT) (Peebles & Xie, 2023). Furthermore, GENREP improves the image generation quality, achieving top-leading performance on CelebA-HQ (Karras et al., 2017), LSUN-Churches (Yu et al., 2015), and ImageNet (Deng et al., 2009) under the class-conditioned setup.

## 2   RELATED WORK

**Diffusion Models for Visual Perception.** Work applying diffusion models for downstream perception tasks can be broadly classified into two categories. The first treats the prediction process as a denoising task, where noisy inputs are refined to recover clean ground truth. This paradigm contains noise-to-box for object/action detection (Chen et al., 2023b; Ho et al., 2023; Nag et al., 2023); noise-to-point for object tracking (Xie et al., 2024b) and pose estimation (Shan et al., 2023; Feng et al., 2023);

and noise-to-map which directly synthesizes colorful masks for depth estimation (Ke et al., 2024), segmentation (Li et al., 2023c; Ji et al., 2023), and anomaly detection (Zhang et al., 2023b). On the other hand, recent research highlights that diffusion models undergoing large-scale pre-training exhibit certain representation abilities, enabling them to extract meaningful features for downstream visual perception tasks (Zhao et al., 2023; Yang & Wang, 2023; Kondapaneni et al., 2024; Chen et al., 2025). On this basis, a significant trend has emerged, where the diffusion models are utilized as backbones for image classification (Clark & Jaini, 2023), image segmentation (Zhu et al., 2024a; Xu et al., 2023), 3D Object Detection (Xu et al., 2024), human-object interaction detection (Li et al., 2024b), and referring video object segmentation (Zhu et al., 2024b). This also facilitates correspondence matching by calculating cosine similarity between diffusion features (Tang et al., 2023; Zhang et al., 2023a). Though demonstrating promising performance, these work often sacrifices the image generation capabilities of models. In contrast, our work seeks to enable both image generation and understanding within the same model, while the distribution knowledge is explicitly transferred from diffusion models to inform and guide the discriminative learning process.

**Joint Discriminative and Generative Learning.** Substantial research has emerged to combine the strengths of both discriminative and generative learning even before the deep learning era. To address the data-intensive and limited generalization inherent in purely discriminative methods, researchers incorporated generative techniques to manage noisy inputs (Jaakkola & Haussler, 1999) and unlabeled samples (Bernardo et al., 2007). Similarly, there are interests in the 'discriminative training' of generative models to mitigate mismatches between real and model-specified data distributions (Tu; Holub & Perona, 2005; Yakhnenko et al., 2005; Chen et al., 2024). More recently, complementary learning methods simultaneously learn data distributions leveraging advanced generative models, such as Generative Adversarial Networks (GANs) (Xu et al., 2020), Variational Autoencoders (VAEs) (Chen et al., 2023a; Kolesnikov et al., 2022), and Gaussian Mixture Models (GMMs) (Liang et al., 2022), resulting in generative classifiers for discriminative tasks. Additionally, generative models are trained to capture the distribution of known classes in open-vocabulary recognition which facilitates the recognition of novel classes (Perera et al., 2020), and tuning diffusion models with a discriminative adapter has proven effective in improving the alignment between text prompts and generated images (Qu et al., 2024). However, most existing work merely focuses on the one-direction enhancement, *e.g.*, discriminative learning to improve image generation or generative learning to enhance visual perception. In contrast, GENREP builds a feedback loop to enable mutual boosts between generative and discriminative learning, while within a unified model.

**Unified Image Understanding and Synthesis.** In recent years, there has been a notable surge in integrating image comprehension and generation within the same model. The first research direction is built upon LLMs, and distinguishes itself by implementing image generation in an auto-regressive manner (Dong et al., 2024), delivering a Tokenizer-Detokenizer framework that enables token-by-token generation of multimodal outputs for synthesis and understanding tasks (Zhu et al., 2023; Ge et al., 2024; Fang et al., 2024; Li et al., 2024a; Wu et al., 2025). Another line of work utilizes diffusion models, which frames perception tasks as the generation of colorful maps (Qi et al., 2024; Wang et al., 2024; Yang et al., 2025) or text embeddings (Huang et al., 2023). Though retaining the generative capability, this kind of solution still falls in low-level reconstruction, lacking high-level modeling on semantics. A notable exception performs discriminative learning using features from diffusion models, and update the generative component in a mean teacher manner (Zheng et al., 2024). However, the image generation capability in this approach is primarily optimized for augmenting perception tasks, leaving its potential for general-purpose image synthesis largely unexplored. To overcome these limitations, GENREP respects and harnesses the unique characteristics of both paradigms. Specifically, it enhances representation learning for perception tasks with generative modeling to consummate the decision boundary, and uses high-level semantics obtained from discriminative learning to instruct the sampling stage (*i.e.*, reverse diffusion) of image synthesis.

## 3 METHODOLOGY

### 3.1 PRELIMINARY: DIFFUSION MODELS FOR VISUAL PERCEPTION

Empirical studies (Zhao et al., 2023; Yu et al., 2024) have demonstrated that features processed by latent diffusion models contain certain visual cues, which can be used to tackle complex perception tasks. Specifically, given an input sample $x$ and its corresponding textual class label $y$, $x$ is first

encoded into the latent space using the encoder $\mathcal{E}$ of a pre-trained generator (*i.e.*, VQGAN), yielding $\boldsymbol{x} = \mathcal{E}(x)$. After a single noise-free forward pass through the denoising network $\epsilon_\theta$ with the encoded label $c_\theta(y)$ as the condition, we obtain $\hat{\boldsymbol{x}} = \epsilon_\theta(\boldsymbol{x}, 0, c_\theta(y))$ which extracts features distinctive for the given class $y$. Following (Zhao et al., 2023), the extracted features are enhanced by aggregating intermediate outputs of four decoder blocks in $\epsilon_\theta$ at different own-sampling factors with FPN(Lin et al., 2017), so as to deliver the final input representations for task-specific decoders. GENREP follows this pipeline to enable downstream visual perception, and further seeks to bridge the historically parallel image generation and understanding tasks, yielding a single model capable of addressing both tasks.

## 3.2 GENREP: RECONCILE VISUAL PERCEPTION AND IMAGE GENERATION

In this section, we first detail how to distill knowledge of visual distributions from diffusion models to enhance discriminative visual perception, and then outline the perception-inspired image generation learning, emphasizing how gained insights from visual perception are utilized to improve generative capabilities. Finally, we address the reconciliation of these dual learning objectives, illustrating how GENREP yields a balanced and unified model proficient in both tasks.

**Generative Visual Perception Learning.** Assuming the diffusion models can capture visual distributions via generative modeling, the conditional distribution for sample $\boldsymbol{x}$ (*i.e.*, $p(\boldsymbol{x}|y)$) can be derived with class label $y$ as the conditional input. Since the exact computation of $p(\boldsymbol{x}|y)$ is intractable, we approximate it following the principle of Markov Chain Monte Carlo (MCMC)(Geyer, 1992). Specifically, we observe that during the reverse diffusion process, a sequence of intermediate states $\boldsymbol{x}_T \rightarrow \boldsymbol{x}_{T-1} \rightarrow \cdots \rightarrow \boldsymbol{x}_0$ naturally constitutes a non-stationary Markov chain(Norris, 1998). The transition kernel $p_\theta(\boldsymbol{x}_{t-1}|\boldsymbol{x}_t)$ at each step can be parameterized as:

$$p_\theta(\boldsymbol{x}_{t-1}|\boldsymbol{x}_t) = \mathcal{N}(\boldsymbol{x}_{t-1}; \mu_\theta(\boldsymbol{x}_t, t), \sigma_\theta(\boldsymbol{x}_t, t)), \tag{1}$$

where $\mathcal{N}$ is a Gaussian distribution, and $\mu_\theta$, $\sigma_\theta$ are networks parameterized by $\theta$ to predict the mean $\mu_t$ and variance $\sigma_t$ for $\mathcal{N}$ at time $t$. This structure is analogous to MCMC methods where samples are drawn from a sequence of transitions rather than independent draws from a static distribution.

However, leveraging the chain directly for approximation introduces two challenges: **i)** the initial states of the reverse chain correspond to nearly pure noise, which would degrade the approximation quality; and **ii)** samples drawn sequentially from a Markov chain are temporally correlated, which conflicts the independent assumption that strengthens Monte Carlo methods. To mitigate these issues, we adopt two techniques, known as ***burn-in*** and ***thinning***, commonly used in MCMC(Link & Eaton, 2012). For the ***burn-in*** period, we discard the first $m$ uninformative steps of the chain. For ***thinning***, we reduce the correlation by selecting samples at a fixed interval of $k$-th step. Empirical evaluations show that $k = 2$ provide a good trade-off between sample quality and quantity. Following the practice of MCMC, we then leverage the trajectory of a single reverse diffusion process to estimate $p(\boldsymbol{x}|y)$:

$$p(\boldsymbol{x}|y) \approx \tfrac{1}{T} \sum_{t=1}^{T} \mathcal{N}(\boldsymbol{x}; \mu_{t,y}, \sigma_{t,y}), \tag{2}$$

where $T$ represents the total number of reverse diffusion steps after burn-in and thinning. This allows for a highly efficient estimation by avoiding the need to generate a large set of fully-denoised samples $\boldsymbol{x}_0$ (*i.e.*, massive full reverse diffusion runs) for each condition $y$, as required by standard Monte Carlo methods. The posterior distribution $p(y|\boldsymbol{x})$ is then computed substitute into the Bayes' theorem:

$$p(y|\boldsymbol{x}) = \frac{p(y)p(\boldsymbol{x}|y)}{\sum_{y' \in \mathcal{Y}} p(y')p(\boldsymbol{x}|y')}, \tag{3}$$

where $p(y) = 1/|\mathcal{Y}|$ is assumed to be uniformly distributed. This is a standard choice(Kingma et al., 2014; Tran et al., 2019) which creates a non-informative prior that allows the posterior distribution to be shaped primarily by the learned likelihood $p(\boldsymbol{x}|y)$, which contains the rich distributional knowledge we aim to distill. While (Li et al., 2023a) also uses diffusion models to estimate conditional distributions with Monte Carlo methods, it approximates $\log p(\boldsymbol{x}|y)$ by averaging the noise prediction error derived from the forward diffusion process. In contrast, this work directly approximates $p(\boldsymbol{x}|y)$ by averaging Gaussian PDF values predicted during the reverse generative process (*i.e.*, Eq. 1). The motivation (*i.e.*, correct conditioning enjoying accurate noise prediction *vs* patterns of samples generated with the same conditions being consist), computational basis (*i.e.*, noise prediction error *vs* Gaussian probability densities), and diffusion process (*i.e.*, forward noising-adding *vs* reverse

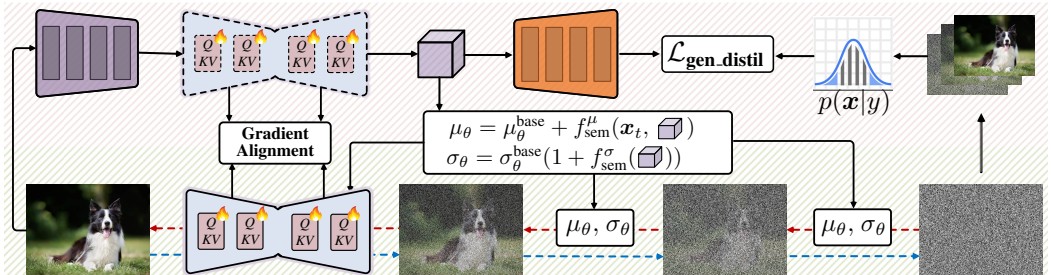

Figure 2: The overall pipeline of GenRep (§3.2). First, our proposed generative visual perception learning (*i.e.*, region on ▨) transfers distribution knowledge from diffusion models, in leverage of intermediate denoised images as samples to approximate the conditional distribution (*i.e.*, $p(\boldsymbol{x}|y)$). Second, semantic-driven generation learning (*i.e.*, region on ▨) utilizes the semantic embeddings (*i.e.*, ▱) learned from visual perception tasks to guide the image generation process. Finally, gradients generated by these two type of losses are aligned via Eq. 16, to deliver a unified model that excels in both image generation and synthesis tasks.

generation) are all different. Our approach requires significantly fewer diffusion steps (*i.e.*, 1000 *vs* 200), and thus excels in computational efficiency.

To inform the discriminative perception process with distributional knowledge, we minimize the Kullback-Leibler (KL) divergence between $p(y|\boldsymbol{x})$ computed by generative modeling and $q(y|\boldsymbol{x})$ obtained by applying the `softmax` operation to the output logits $\boldsymbol{z}$ of task-specific decoders:

$$\mathcal{L}_{\text{gen\_distil}} = D_{\text{KL}}(p||q) = \sum_{y \in \mathcal{Y}} p(y|\boldsymbol{x}) \log \frac{p(y|\boldsymbol{x})}{q(y|\boldsymbol{x})}. \tag{4}$$

The final objective combines $\mathcal{L}_{\text{gen\_distil}}$ with the conventional discriminative loss $\mathcal{L}_{\text{disc}}$ for each perception task (*e.g.*, Cross Entropy loss for classification, Smooth $\ell_1$ loss for bounding box regression):

$$\mathcal{L}_{\text{percept}} = \mathcal{L}_{\text{disc}} + \mathcal{L}_{\text{gen\_distil}}. \tag{5}$$

Eq. 5 bridges the gap between generative and discriminative frameworks with $\mathcal{L}_{\text{gen\_distil}}$ as a regularizer. Unlike standard discriminative loss that encourages overconfident predictions, $\mathcal{L}_{\text{gen\_distil}}$ leverages the generative likelihood to create a soft posterior that faithfully reflects ambiguity for similar classes.

**Semantic-Driven Generation Learning.** To enhance image generation, we propose a semantic-aware noise adjustment strategy which leverages high-level semantics learned through visual perception. Assuming there is a well-trained denoising network optimized for visual perception (*i.e.*, $\mathcal{L}_{\text{percept}}$ in Eq. 5), the intermediate output representation which contains rich semantic cues is denoted as $\boldsymbol{x}_{\text{sem}}$. During the reverse diffusion process, the noise parameters (*i.e.*, mean $\mu_\theta$ and variance $\sigma_\theta$ in Eq. 1) are dynamically modulated according to $\boldsymbol{x}_{\text{sem}}$. Specifically, the prediction for $\mu_\theta$ is augmented as:

$$\mu_\theta(\boldsymbol{x}_t, t, \boldsymbol{x}_{\text{sem}}) = \mu_\theta^{\text{base}}(\boldsymbol{x}_t, t) + f_{\text{sem}}^\mu(\boldsymbol{x}_t, \boldsymbol{x}_{\text{sem}}), \tag{6}$$

where $\mu_\theta^{\text{base}}(\boldsymbol{x}_t, t)$ is the baseline mean value predicted by the underlying diffusion model at time step $t$ and $f_{\text{sem}}^\mu$ is a semantic correction function implemented as:

$$f_{\text{sem}}^\mu(\boldsymbol{x}_t, \boldsymbol{x}_{\text{sem}}) = \boldsymbol{W}_t^\mu \cdot \text{concat}(\boldsymbol{x}_t, \boldsymbol{x}_{\text{sem}}), \tag{7}$$

with $\boldsymbol{W}_t^\mu$ being a learned weight matrix. Conceptually, $\mu_\theta$ determines the primary denoising direction, and adjusts it by steering the denoising trajectory towards the desired semantic target encoded in $\boldsymbol{x}_{\text{sem}}$. On the other hand, the variance $\sigma_\theta$ controls the uncertainty of reverse diffusion. Dynamically modulating $\sigma_\theta$ allows the model to adaptively control the influence of semantic guidance. The variance prediction network is correspondingly augmented as:

$$\sigma_\theta(\boldsymbol{x}_t, t, \boldsymbol{x}_{\text{sem}}) = \sigma_\theta^{\text{base}}(\boldsymbol{x}_t, t) \cdot (1 + f_{\text{sem}}^\sigma(\boldsymbol{x}_{\text{sem}})), \tag{8}$$

where $\sigma_\theta^{\text{base}}(\boldsymbol{x}_t, t)$ is the baseline variance predicted following the pipeline of improved DDPM(Nichol & Dhariwal, 2021), and $f_{\text{sem}}^\sigma(\boldsymbol{x}_{\text{sem}})$ is a semantic scaling factor computed as:

$$f_{\text{sem}}^\sigma(\boldsymbol{x}_{\text{sem}}) = \text{MLP}(\boldsymbol{x}_{\text{sem}}), \tag{9}$$

where MLP maps the semantic embedding to a scalar. Intuitively, a positive $f_{\text{sem}}^\sigma(\boldsymbol{x}_{\text{sem}})$ increases variance, encouraging broader exploration towards the semantic target when the current sample is far.

Conversely, a negative value reduces variance, promoting finer refinement near the target semantics. The overall training objective combines the standard reconstruction loss for latent diffusion models (*i.e.*, $\mathcal{L}_{\text{LDM}}$) and the representation alignment loss (*i.e.*, $\mathcal{L}_{\text{rep\_align}}$):

$$\mathcal{L}_{\text{genera}} = \mathcal{L}_{\text{LDM}} + \mathcal{L}_{\text{rep\_align}}, \tag{10}$$

where $\mathcal{L}_{\text{rep\_align}}$ minimizes the cosine similarity between $x$ and $x_{\text{sem}}$ as in (Yu et al., 2024). During inference, the explicit semantic representation $x_{\text{sem}}$ in Eq. 6 and Eq. 8 can be directly replaced with the current noisy sample $x_t$, as the enhanced denoising network has already learned to capture necessary semantic cues with the knowledge preserved in model weights.

**Gradient Alignment for Weight Merge.** To reconcile the optimization of visual perception loss (*i.e.*, $\mathcal{L}_{\text{percept}}$ in Eq. 5) and image generation loss (*i.e.*, $\mathcal{L}_{\text{genera}}$ in Eq. 10) within a single model, a gradient alignment mechanism is introduced to address potential conflicts between the two training objectives by symmetrically modifying their respective gradients according to the severity of the conflict. Let $\nabla\mathcal{L}_{\text{percept}}$ and $\nabla\mathcal{L}_{\text{genera}}$ denote the gradients derived from $\mathcal{L}_{\text{percept}}$ and $\mathcal{L}_{\text{genera}}$, respectively. We decompose each gradient into components parallel and orthogonal to the other gradient:

$$\nabla\mathcal{L}_{\text{genera}}^{\|} = \frac{\nabla\mathcal{L}_{\text{percept}} \cdot \nabla\mathcal{L}_{\text{genera}}}{\|\nabla\mathcal{L}_{\text{percept}}\|^2}\nabla\mathcal{L}_{\text{percept}}, \quad \nabla\mathcal{L}_{\text{genera}}^{\perp} = \nabla\mathcal{L}_{\text{genera}} - \nabla\mathcal{L}_{\text{genera}}^{\|}, \tag{11}$$

$$\nabla\mathcal{L}_{\text{percept}}^{\|} = \frac{\nabla\mathcal{L}_{\text{percept}} \cdot \nabla\mathcal{L}_{\text{genera}}}{\|\nabla\mathcal{L}_{\text{genera}}\|^2}\nabla\mathcal{L}_{\text{genera}}, \quad \nabla\mathcal{L}_{\text{percept}}^{\perp} = \nabla\mathcal{L}_{\text{percept}} - \nabla\mathcal{L}_{\text{percept}}^{\|}. \tag{12}$$

Here the parallel components capture movements in the same or opposite gradient directions of two tasks, while the orthogonal components are gradient directions that do not affect the objective of the other task (Farajtabar et al., 2020). The aligned gradients for both tasks are then reconstructed as:

$$\nabla_{\text{genera}}^{\text{aligned}} = \nabla\mathcal{L}_{\text{genera}}^{\perp} + \alpha\nabla\mathcal{L}_{\text{genera}}^{\|}, \tag{13}$$

$$\nabla_{\text{percept}}^{\text{aligned}} = \nabla\mathcal{L}_{\text{percept}}^{\perp} + \alpha\nabla\mathcal{L}_{\text{percept}}^{\|}. \tag{14}$$

This approach selectively dampens gradient components parallel to the other, while fully preserving the orthogonal ones. Consequently, non-conflicting information is retained, and interference is smoothly reduced based on the conflict level. Here $\alpha$ is a adaptive retention factor governing the damping and defined according to the cosine similarity between two original gradients:

$$\text{cos\_sim} = \frac{\nabla\mathcal{L}_{\text{percept}} \cdot \nabla\mathcal{L}_{\text{genera}}}{\|\nabla\mathcal{L}_{\text{percept}}\|\|\nabla\mathcal{L}_{\text{genera}}\|}. \tag{15}$$

We want $\alpha = 1$ when $\text{cos\_sim} = 1$ (no damping needed) and $\alpha$ to decrease towards 0 as $\text{cos\_sim} \rightarrow -1$ (maximum damping). A simple and effective formulation is the scaled and shifted power function: $\alpha = ((\text{cos\_sim} + 1)/2)^k$. Here, $k = 2$ is a hyperparameter controlling the sharpness of the damping. The final gradients used for the model update are a weighted sum of aligned gradients:

$$\nabla_{\text{symmetric}}^{\text{aligned}} = w_p\nabla_{\text{percept}}^{\text{aligned}} + w_g\nabla_{\text{genera}}^{\text{aligned}}, \tag{16}$$

where $w_p = 0.7$ and $w_g = 0.3$ scale task weights. As such, GENREP effectively manages gradient conflicts during joint learning, and encourages balanced optimization across two objectives.

### 3.3 IMPLEMENTATION DETAILS

**Network Architecture.** GENREP is built upon LDM-8 (Rombach et al., 2022)/DiT-XL (Peebles & Xie, 2023) with 200/250 DDPM steps during inference. To ensure fair comparisons with existing work, the diffusion model is initialized with weights pretrained on ImageNet (Deng et al., 2009) and the LAION dataset (Schuhmann et al., 2022; 2021), respectively. This facilitates comparison against conventional *discriminative-based perception models* pretrained on ImageNet-1K, and other *diffusion-based perception approaches* pretrained on large-scale image-text pairs. Task-specific decoders are designed following representative work with details provided in *Appendix*.

**Training Strategy.** GENREP is first optimized with solely task-specific perception loss ($\mathcal{L}_{\text{percept}}$, Eq. 5), yielding in denoising network $\epsilon_\theta^{\text{sem}}$ which encodes high-level semantic cues into the intermediate output $x_t$ (resulting in $x_{\text{sem}}$). Subsequently, the images generation loss ($\mathcal{L}_{\text{genera}}$, Eq. 10) steps in. A new denoising network $\epsilon_\theta^{\text{unified}}$ copied from $\epsilon_\theta^{\text{sem}}$ is optimized where at each training step:

Table 1: Quantitative results for fine-grained bird classification on CUB-200(Wah et al., 2011) `test` and OOD generalization on ObjectNet(Barbu et al., 2019) `test`.

| Model | Pre-Training | CUB-200 | ObjectNet |
|---|---|---|---|
| ResNet-50 (He et al., 2016) | ImageNet | 84.5 | 37.2 |
| Swin-S (Liu et al., 2021) | ImageNet | 88.2 | 38.9 |
| ConvNeXt-S (Liu et al., 2022) | ImageNet | 88.5 | 39.5 |
| HorNet-S (Rao et al., 2022) | ImageNet | 89.1 | 39.3 |
| GENREP$_{LDM}$ | ImageNet | **90.5** | **51.1** |
| Swin-B (Liu et al., 2021) | ImageNet | 90.6 | 40.3 |
| ConvNeXt-B (Liu et al., 2022) | ImageNet | 90.9 | 40.9 |
| HorNet-B (Rao et al., 2022) | ImageNet | 91.2 | 40.6 |
| GENREP$_{DiT}$ | ImageNet | **92.1** | **54.7** |
| Clark et al. (Clark & Jaini, 2023) | LAION-5B | 91.5 | 49.4 |
| Li et al. (Li et al., 2023a) | LAION-5B | 91.8 | 52.5 |
| GENREP$_{LDM}$ | LAION-5B | **92.9** | **57.8** |

Table 2: Quantitative results for monocular depth estimation on NYUv2(Silberman et al., 2012) `val`.

| Model | Pre-Training | $\delta_1 \uparrow$ | $\delta_3 \uparrow$ | AbsRel $\downarrow$ |
|---|---|---|---|---|
| BTS(Lee et al., 2019) | ImageNet | 0.882 | 0.996 | 0.108 |
| P3Depth(Patil et al., 2022) | ImageNet | 0.898 | 0.996 | 0.104 |
| TransDepth(Zhao et al., 2021) | ImageNet | 0.900 | 0.996 | 0.106 |
| AdaBins(Bhat et al., 2021) | ImageNet | 0.903 | 0.997 | 0.103 |
| DPT(Ranftl et al., 2021) | ImageNet | 0.904 | 0.998 | 0.110 |
| BinsFormer(Li et al., 2024c) | ImageNet | 0.925 | 0.997 | 0.094 |
| ZoeDepth(Bhat et al., 2023) | ImageNet | 0.951 | 0.999 | 0.077 |
| GENREP$_{LDM}$ | ImageNet | **0.964** | **0.999** | **0.070** |
| GENREP$_{DiT}$ | ImageNet | **0.968** | **0.999** | **0.064** |
| VPD(Zhao et al., 2023) | LAION-5B | 0.964 | 0.999 | 0.069 |
| ECoDepth(Patni et al., 2024) | LAION-5B | 0.978 | 0.997 | 0.059 |
| DepthAnything(Yang et al., 2024) | 62M Depth | 0.984 | 1.000 | 0.056 |
| GENREP$_{LDM}$ | LAION-5B | **0.982** | **1.000** | **0.057** |

- Gradients for both $\mathcal{L}_{percept}$ and $\mathcal{L}_{genera}$ are computed using the same input image;
- Gradients are aligned according to Eq. 16 to update weights of attention blocks in $\epsilon_\theta^{unified}$;
- Parameters of $\epsilon_\theta^{sem}$ are updated in a momentum manner: $\theta^{sem} \leftarrow m\theta^{sem} + (1-m)\theta^{unified}$ with $m = 0.999$. This maintains stable semantic features $\boldsymbol{x}_{sem}$ for image generation learning.

## 4 EXPERIMENT

**Datasets.** The experiments are conducted on nine datasets. Concretely, CUB-200 (Wah et al., 2011) for fine-grained bird classification, ObjectNet(Barbu et al., 2019) for out-of-the-distribution generation, NYUv2(Silberman et al., 2012) for depth estimation, ADE20K(Zhou et al., 2017) for open/close set semantic segmentation, MS COCO(Lin et al., 2014) and LViS v1.0(Gupta et al., 2019) for open-vocabulary object detection, ImageNet(Deng et al., 2009), CelebA-HQ(Karras et al., 2017), and LSUN-Churches(Yu et al., 2015) for image generation. Details are provided in *Appendix*.

**Evaluation Metrics.** For fine-grained classification on CUB-200 and out-of-the-distribution generalization on ObjectNet, we report the top-1 accuracy. For depth estimation, following (Li et al., 2024c), we report the accuracy under the threshold ($\delta_i < 1.25^i, i = 1, 3$) and mean absolute relative error (AbsRel). For close-set and open-vocabulary semantic segmentation, following (Xu et al., 2022a; Cho et al., 2024), GENREP is trained on the training set of ADE20K and COCO Stuff, respectively. The evaluation is conducted on the validation set of ADE20K with the mIoU score reported. For open-vocabulary object detection, consistent with prior work (Zang et al., 2022; Wu et al., 2023a), we report the $AP_{50}$ score for base, novel, and all classes, denoted as $AP_{50}^b$, $AP_{50}^n$, $AP_{50}$ on MS COCO, $AP_r$, $AP_c$, $AP_f$, and AP for rare (novel), common, frequent, and all categories on LVIS. For image generation, following (Rombach et al., 2022), we report the FID, IS, precision, and recall scores.

**Training.** For visual classification, we use standard data augmentation techniques, including random cropping and horizontal flipping during training to enhance generalization. The AdamW optimizer with a learning rate of $1e^{-3}$ and a weight decay of 0.05 is adopted. The batch size is set to 256 with 50 epochs training. For depth estimation, following (Li et al., 2024c), we train the model for 40K steps with a batch size of 16, and use the Adam optimizer with a learning rate of $1e^{-4}$ and a weight decay of $5e^{-2}$. For semantic segmentation, following (Cheng et al., 2022; Xie et al., 2024a; Li et al., 2023b), the model is optimized with AdamW using a learning rate of $2e^{-4}$ and a weight decay of $1e^{-4}$ for 80K iterations on COCO Stuff for open-vocabulary, and 160K iterations on ADE20K for close-set. Input images are cropped to the $768 \times 768$ pixels. For open-vocabulary object detection, following (Zhao et al., 2024; Zhang et al., 2024), we train GENREP for 40K steps on MS COCO and 80K steps on LViS v1.0 with a batch size of 16, and adopt the Adam optimizer with a learning rate of $2e^{-3}$ and a weight decay of $1e^{-4}$. Given the simultaneous training of both perception and generation in GENREP, the training procedure for image synthesis is aligned with perception tasks.

### 4.1 COMPARISON WITH STATE-OF-THE-ARTS

**Visual Recognition.** As shown in Table 1, benefited from the low-level modeling ability of diffusion models, GENREP yields remarkable performance on the bird classification task which prioritizes fine-grained cues. Furthermore, the knowledge transfer from diffusion models allows GENREP to achieve a top-1 accuracy of **54.7%/57.8%** for out-of-distribution generalization on ObjectNet, surpassing prior diffusion-based methods(Clark & Jaini, 2023; Li et al., 2023a) by **8.4%/5.3%**.

Table 3: Quantitative results for closed-set semantic segmentation on ADE20K(Zhou et al., 2017) `val`.

| Model | Pre-Training | Backbone | mIoU ↑ |
|---|---|---|---|
| DeepLabV3+ (Chen et al., 2018) | ImageNet | ResNet-101 | 45.5 |
| OCRNet (Yuan et al., 2020) | ImageNet | HRNet-W48 | 45.7 |
| UperNet (Xiao et al., 2018) | ImageNet | Swin-S | 47.7 |
| SegMentor (Strudel et al., 2021) | ImageNet | DeiT-B | 47.1 |
| K-Net (Zhang et al., 2021) | ImageNet | Swin-S | 49.7 |
| SegFormer (Xie et al., 2021) | ImageNet | MiT-B5 | 50.0 |
| Mask2Former (Cheng et al., 2022) | ImageNet | Swin-S | 51.3 |
| GENREP | ImageNet | LDM | 52.2 |
| GENREP | ImageNet | DiT | 52.8 |
| SDN (Tan et al., 2022) | LAION-5B | LDM | 51.1 |
| VPD (Zhao et al., 2023) | LAION-5B | LDM | 53.7 |
| GENREP | LAION-5B | LDM | **54.6** |

Table 4: Quantitative results for open-vocabulary semantic segmentation on ADE20K(Zhou et al., 2017) `val`.

| Model | Pre-Training | Backbone | mIoU ↑ |
|---|---|---|---|
| GroupViT (Xu et al., 2022a) | ImageNet | ViT-S | 10.6 |
| ZegFormer (Ding et al., 2022) | ImageNet | ViT-B | 18.0 |
| SimBaseline (Xu et al., 2022b) | ImageNet | ViT-B | 20.5 |
| PACL (Mukhoti et al., 2023) | ImageNet | ViT-B | 31.4 |
| OVSeg (Liang et al., 2023) | ImageNet | ViT-B | 24.8 |
| CAT-Seg (Cho et al., 2024) | ImageNet | ViT-B | 27.2 |
| SED (Xie et al., 2024a) | ImageNet | ConvNeXt-B | 31.6 |
| GENREP | ImageNet | LDM | **32.5** |
| GENREP | ImageNet | DiT | **34.1** |
| OVDiff (Karazija et al., 2024) | LAION-5B | LDM | 14.1 |
| ODISE (Xu et al., 2023) | LAION-5B | LDM | 28.7 |
| GENREP | LAION-5B | LDM | **34.7** |

Table 5: Open-vocabulary detection on MS COCO(Lin et al., 2014) and LVIS v1.0(Gupta et al., 2019) `val`.

| Model | Visual-Linguistic Models | MS COCO | | | LViS v1.0 | | | |
|---|---|---|---|---|---|---|---|---|
| | | $AP_{50}^n$ ↑ | $AP_{50}^b$ ↑ | $AP_{50}$ ↑ | $AP_r$ ↑ | $AP_c$ ↑ | $AP_f$ ↑ | $AP$ ↑ |
| ViLD(Gu et al., 2022) | CLIP | 27.6 | 59.9 | 51.2 | 16.1 | 20.0 | 28.3 | 22.5 |
| OV-DETR(Zang et al., 2022) | CLIP | 29.4 | 61.0 | 52.7 | 17.4 | 25.0 | 32.5 | 26.6 |
| OADP(Wang et al., 2023) | CLIP | 35.6 | 55.8 | 50.5 | 19.9 | 26.0 | 28.7 | 26.0 |
| BARON(Wu et al., 2023a) | CLIP | 34.0 | 60.4 | 53.5 | 23.2 | 29.3 | 32.5 | 29.5 |
| CORA(Wu et al., 2023b) | CLIP | 35.1 | 35.5 | 35.4 | 28.1 | - | - | - |
| BIND(Zhang et al., 2024) | CLIP | 36.3 | 54.7 | 50.2 | 29.4 | 30.6 | 33.5 | 31.4 |
| SAS-Det(Zhao et al., 2024) | CLIP | 37.4 | 58.5 | 53.0 | 29.1 | 32.4 | 36.8 | 33.5 |
| GENREP | LDM | **41.8** | 60.8 | 55.1 | 30.5 | 33.3 | 35.8 | 34.8 |
| GENREP | DiT | **43.4** | 61.5 | 56.5 | 31.6 | 33.7 | 37.3 | 36.0 |

Table 6: Quantitative results for class-conditional image generation on ImageNet(Deng et al., 2009) 256×256.

| Model | FID↓ | IS↑ | Precision ↑ | Recall↑ |
|---|---|---|---|---|
| BigGAN(Brock et al., 2018) | 6.95 | 171.4 | 0.87 | 0.28 |
| StyleGAN(Karras et al., 2021) | 2.30 | 265.1 | 0.78 | 0.53 |
| ADM(Dhariwal & Nichol, 2021) | 4.59 | 186.7 | 0.82 | 0.52 |
| CDM(Ho et al., 2022) | 4.88 | 158.7 | - | - |
| RIN(Jabri et al., 2023) | 3.42 | 182.0 | - | - |
| VDM++(Kingma & Gao, 2023) | 2.12 | 267.7 | - | - |
| LDM-8(Rombach et al., 2022) | 7.77 | 201.6 | 0.84 | 0.35 |
| +GENREP | 6.92 | 213.7 | 0.89 | 0.44 |
| DiT-XL(Peebles & Xie, 2023) | 2.27 | 278.2 | 0.83 | 0.57 |
| +GENREP | 2.09 | 283.8 | 0.88 | 0.58 |

Table 7: Quantitative results for image generation on CelebA-HQ and LSUN-Churches 256×256.

| Model | FID↓ | Precision ↑ | Recall↑ |
|---|---|---|---|
| ***CelebA-HQ*** | | | |
| PGGAN(Karras et al., 2017) | 8.0 | - | - |
| UDM(Meng et al., 2022) | 7.16 | - | - |
| LDM-4(Rombach et al., 2022) | 5.11 | 0.72 | 0.49 |
| +GENREP | 3.84 | 0.78 | 0.54 |
| ***LSUN-Churches*** | | | |
| PGGAN(Karras et al., 2017) | 6.42 | - | - |
| StyleGAN(Karras et al., 2019) | 4.21 | - | - |
| LDM-8(Rombach et al., 2022) | 4.02 | 0.64 | 0.52 |
| +GENREP | 3.12 | 0.69 | 0.58 |

**Depth Estimation.** For depth estimation, as shown in Table 2, GENREP achieves an impressive score of **0.064** in term of AbsRel. This verifies our core design to conduct both generative and discriminative learning. Moreover, after initializing weights from Stable Diffusion pretraining on LAION-5B(Schuhmann et al., 2022), GENREP achieves comparable performance to DepthAnything(Yang et al., 2024) which is pretrained on 1.5M labeled and 62M unlabeled depth samples.

**Semantic Segmentation.** A detailed comparison of GENREP against top-leading approaches for semantic segmentation is provided in Tables 3-4. Built upon LDM(Rombach et al., 2022), GENREP achieves a **52.2%/54.6%** mIoU for close-set semantic segmentation on ADE20K, beating all competitors. Moreover, for open-vocabulary semantic segmentation, our method delivers a **6.0%** gain over ODISE (Xu et al., 2023). Leveraging DiT(Peebles & Xie, 2023) as the backbone observes similar trends, and builds new SOTA on two setups.

**Object Detection.** As shown in Table 5, GENREP demonstrates remarkable accuracy over existing work for open-vocabulary object detection on MS COCO (*e.g.*, **41.8%** *v.s.* 37.4% in terms of $AP_{50}^n$), and LViS v1.0 (*e.g.*, **30.5%** *v.s.* 29.1% in terms of $AP_r$). When using the Transformer-based diffusion models (*i.e.*, DiT(Peebles & Xie, 2023)), the performance boosts to **43.4%** $AP_{50}^n$ and **31.6%** $AP_r$.

**Image Generation.** Image generation results on ImageNet(Deng et al., 2009), CelebA-HQ(Karras et al., 2017), and LSUN-Churches(Yu et al., 2015) are presented in Tables 6-7. As seen, GENREP boosts the performance to new SOTAs across metrics, proving the effectiveness of the overall design.

## 4.2 QUALITATIVE RESULTS

Fig.3 presents visualization results for visual perception on ADE20K, NYUv2, MS COCO, and for image generation on ImageNet, CelebA-HQ, LSUN-Churches. It can be observed that GENREP could effectively handle challenging scenarios, while synthesizing high-quality images.

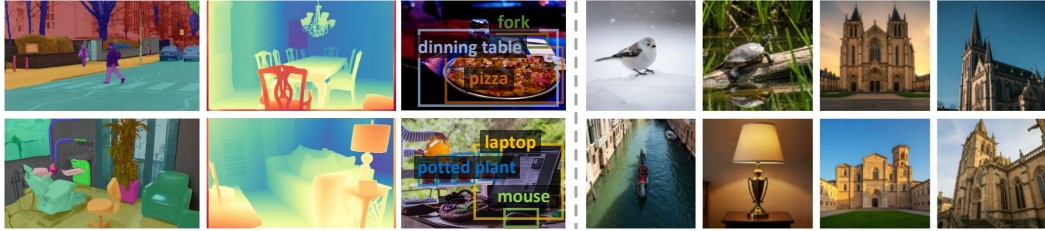

Figure 3: Visualization results for image understanding on ADE20K(Zhou et al., 2017), NYUv2(Silberman et al., 2012), MS COCO(Lin et al., 2014), and for image generation on ImageNet(Deng et al., 2009), LSUN-Churches(Yu et al., 2015).

Table 8: Analysis of essential components in GENREP.

| Generative Vis. Perception | Semantic-Dri. Generation | Gradient Align. | Top-1↑ | mIoU↑ | FID↓ |
|---|---|---|---|---|---|
| | | | 45.4 | 27.8 | 13.27 |
| ✓ | | | 47.8 | 30.9 | 12.96 |
| | ✓ | | 44.1 | 25.6 | 7.45 |
| ✓ | ✓ | | 49.4 | 31.5 | 7.23 |
| ✓ | ✓ | ✓ | **51.1** | **32.5** | **6.92** |

Table 9: Analysis of the thinning interval $k$.

| Interval $k$ | ObjectNet | CUB-200 | ADE20K | MS COCO |
|---|---|---|---|---|
| 1 | 50.3 | 89.2 | 30.7 | 53.5 |
| **2** | **51.1** | **90.5** | **32.5** | **55.1** |
| 3 | 51.3 | 90.7 | 31.8 | 54.6 |
| 4 | 50.5 | 90.3 | 31.5 | 53.5 |
| 5 | 48.9 | 90.0 | 30.9 | 52.2 |

Table 10: Analysis of burn-in sample number $m$.

| $m$ | 25 | **50** | 75 | 100 | 125 |
|---|---|---|---|---|---|
| ObjectNet | 50.4 | **51.1** | 50.2 | 48.8 | 47.6 |
| ADE20K | 32.3 | **32.5** | 32.1 | 31.3 | 29.2 |

Table 11: Analysis of expected calibration error (ECE).

| $\mathcal{L}_{gen\_distil}$ | ObjectNet | CUB-200 | ADE20K | MS COCO |
|---|---|---|---|---|
| | 0.237 | 0.095 | 0.484 | 0.382 |
| ✓ | 0.208 | 0.076 | 0.425 | 0.343 |

Table 12: Strategies for semantic-driven generation.

| Noise Adjust. | $\mathcal{L}_{rep\_align}$ | Top-1↑ | mIoU↑ | FID↓ |
|---|---|---|---|---|
| ✓ | | 49.6 | 30.9 | 7.16 |
| | ✓ | 50.3 | 31.5 | 7.38 |
| ✓ | ✓ | **51.1** | **32.5** | **6.92** |

Table 13: Analysis of directions for gradient alignment.

| Gradient Align. | Top-1↑ | mIoU↑ | FID↓ |
|---|---|---|---|
| $\nabla_{genera}^{aligned}$ in Eq. 13 | 48.7 | 30.3 | 6.79 |
| $\nabla_{symmetric}^{aligned}$ in Eq. 16 | **50.1** | **32.5** | **6.92** |

Table 14: Analysis of feature robustness on ObjectNet.

| Model | t=0 (Clean) | t=10 | t=20 | t=50 |
|---|---|---|---|---|
| Swin-Transformer | 40.3 | 23.1 | 11.5 | 4.6 |
| **GENREP** | **51.1** | **48.7** | **44.5** | **37.2** |

## 4.3 DIAGNOSTIC EXPERIMENTS

For in-depth analysis, we conduct ablative studies with LDM(Rombach et al., 2022) as the denoising network. Unless otherwise specified, all experiments use GENREP$_{LDM}$ pretrained on ImageNet.

**Key Component Analysis.** We investigate the essential designs of GENREP, *i.e.*, generative visual perception learning, semantic-driven generation learning, and gradient alignment for weight merge in §3.2 in Table 8. First, our generative visual perception learning strategy proves to be broadly effective across visual perception tasks, yielding notable performance improvements. Second, with semantic-driven generation learning, GENREP delivers promising gains for the image generation task. Third, after combining them (*i.e.*, *row* #3), both image generation and understanding tasks enjoy further boosts, which reveals a positive feedback loop is established. Finally, with gradient alignment to unify the optimization direction, GENREP achieves the best performance on all three datasets.

**Thinning Interval.** We analyze the impact of varying thinning intervals $k$ for MCMC approximation in Table 9. As seen, setting $k = 1$, *i.e.*, using all intermediate samples for approximation yields a moderate improvement over the baseline (*row* #1 in Table 8). When $k = 2$, GENREP enjoys large performance gain. However, further increasing $k$ leads to a decline in performance. This is because a larger $k$ reduces the number of available samples and leads to a high-variance distributional estimate, indicating the trade-off between inference efficiency with fewer samples and approximation accuracy.

**Burn-in Phase.** We examines the impact of dsicarding first $m$ samples during reverse diffusion that are heavily noised (*i.e.*, the burn-in strategy in standard MCMC) in Table 10, with the thinning interval $k = 2$. Empirically, we find that $m = 50$ provides a favorable balance, which removes sufficiently noisy initial samples while retaining enough samples to support reliable estimation.

**Confidence Calibration.** We evaluate the expected calibration error (ECE) for predictions output by discriminative visual perception heads. ECE quantifies the alignment between predicted probabilities and the true likelihood of outcomes, serving as a crucial metric for assessing model reliability. As

Table 15: Runtime comparison of closed-set semantic segmentation models on ADE20K `val`.

| Method | Backbone | Trainable Params (M) | Training Time (GPU Hours) | Inference Speed (FPS) | mIoU |
|---|---|---|---|---|---|
| DeepLabV3+(Chen et al., 2018) | ResNet-101 | 63 | 83 | 14.2 | 45.5 |
| SETR(Zheng et al., 2021) | ViT-L | 308 | 623 | 9.7 | 46.2 |
| UperNet(Xiao et al., 2018) | Swin-S | 81 | 104 | 15.2 | 47.7 |
| MaskFormer(Cheng et al., 2021) | Swin-S | 63 | 53 | 19.6 | 49.8 |
| GENREP (perception only) | LDM | 54 | 79 | 12.6 | 49.3 |
| GENREP | LDM | 54 | 87 | 12.6 | **52.2** |

shown in Table 11, the incorporation of generative distillation loss (*i.e.*, $\mathcal{L}_{\text{gen\_distil}}$ in Eq. 4) leads to a substantial reduction in ECE. This also indicates distribution knowledge in diffusion models can be effectively transferred with $\mathcal{L}_{\text{gen\_distil}}$ to improve the reliability of discriminative models.

**Semantic-Driven Generation.** We examine the impact of semantic-aware noise adjustment and representation alignment (*i.e.*, $\mathcal{L}_{\text{rep\_align}}$) in Table 12. The results demonstrate that both techniques independently contribute to improved generation quality. After combining them together, the FID score shows a significant improvement, highlighting the complementary nature of these two designs.

**Gradient Alignment.** We probe different gradient alignment strategies in Table 13. As seen, while projecting perception loss in the direction of generation loss (*i.e.*, $\nabla_{\text{percept}}^{\text{aligned}}$) obtains better image generation performance, there is a significant drop in perception performance. After balancing the trading off, we adopt a symmetric strategy which treats both tasks equally during conflict resolution (*i.e.*, $\nabla\mathcal{L}_{\text{symmetric}}^{\text{aligned}}$) and performs better in perception tasks while maintaining good generation quality.

**Representation Robustness.** To probe whether GENREP preserves good representation capabilities under noisy inputs, we provide it with latents corrupted by $t = 10$, $t = 20$, and $t = 50$ forward diffusion steps. The results summarized in Table 14 offer empirical evidence for the robustness of learned representations, which stems directly from our model design. The perception module uses the denoising network as the backbone, which is trained to extract semantic structure from noisy input, and remains effective when operating on corrupted latents. Furthermore, the conditional distribution $p(\boldsymbol{x}|y)$ for knowledge distillation aggregates noised states throughout the reverse diffusion. This encourages the model to learn noise-tolerant features that are predictive of the correct semantic labels.

## 4.4 RUNTIME ANALYSIS

We present a detailed runtime analysis in Table 15. It is important to emphasize that GENREP is designed as a truly unified model that simultaneously masters visual perception and image generation within a single training process. The competitors, in contrast, are optimized exclusively for segmentation. From this unified perspective, the efficiency of GENREP is remarkable. With a total training cost of 87 GPU hours, it not only learns a strong image generator but also delivers a SOTA perception model that achieves 52.2% mIoU, surpassing all listed specialist models. To further isolate the cost of our proposed generative distillation, we compare the full GENREP to a perception-only variant that removes the MCMC-based approximation. As seen, the full model incurs a modest 10.1% increase in training time, yet elevates mIoU by a significant of 2.9 points.

## 5 CONCLUSION

In this work, we reconcile visual perception and image generation within a unified model, termed GENREP. This leads to joint discriminative and generative learning, where the unique properties of both paradigms are preserved and utilized to enhance each other. To achieve an optimal state for both image understanding and synthesis tasks, a gradient alignment strategy is proposed to pull close the weights optimized for two tasks. Empirical results suggest that GENREP achieves superior performance on six perception benchmarks, and greatly improves the image generation ability. Beyond the strong empirical results, our framework naturally inherits the flexible multimodal conditioning capabilities from LDM. This positions GENREP as a promising foundation to reconcile multimodal understanding and generation in one unified model, a key direction for future work.

**Acknowledgement.** This work was supported by National Science and Technology Major Project (No. 2023ZD0121300), National Natural Science Foundation of China (No. 62372405), Zhejiang Provincial Natural Science Foundation of China (No. LD25F020001), Beijing Natural Science Foundation (L252036), Fundamental Research Funds for the Central Universities (226-2025-00057), Jiangsu Provincial Scientific Research Center of Applied Mathematics (No. BK20233002), and CCF-Tencent Open Fund.

**Ethics Statement.** This paper explores the reconciliation of visual perception and generation in diffusion models. It does not introduce new ethical concerns beyond those well established in the community. We do not identify any specific risks that warrant ethical review. For the potential misuse in deepfake generation, we encourage responsible deployment and support discussions on policy and regulatory frameworks to ensure the ethical application of generative models.

**Reproducibility.** GENREP is implemented in PyTorch and trained on four Tesla A100 GPUs. Testing is carried on the same machine. Our code is available at GENREP.

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

# A APPENDIX

## A.1 DECLARATION OF LLM USAGE

The LLM was used solely for grammar checking and did not contribute to the core methodological design or the originality of the research.

## A.2 LIMITATIONS

One potential limitation of GENREP is its computational cost, which introduces a trade-off between model performance and inference efficiency. Our reliance on a diffusion model backbone results in a lower inference speed compared to highly specialized perception architectures. As detailed in Table 15, GENREP operates at 12.6 FPS for semantic segmentation, whereas models like MaskFormer achieves 19.6 FPS and DeepLabV3+ achieves 14.2 FPS. This may constrain the usage of GENREP in latency-sensitive applications, such as real-time analysis or autonomous systems. This trade-off is motivated by the substantial benefits our unified approach provides, including a 2.3% mIoU improvement over MaskFormer and, crucially, stronger generalization to out-of-distribution data. Our work aligns with the growing trend of using large-scale generative models to unlock new capabilities in visual understanding, which often involves an initial focus on performance over efficiency. We consider the optimization of unified models a vital direction for future research. Promising avenues include knowledge distillation to yield lightweight architectures, developing more efficient diffusion sampling techniques tailored for perception, and model quantization. Bridging this efficiency gap will benefit the deployment of powerful unified perception and generation models in practical scenarios.

Furthermore, in diffusion models, the mean of the data distribution is far more dominant than the variance (Nichol & Dhariwal, 2021). Consequently, the learned variance can be less precise. Our method mitigates this by considering $p(x|y)$ as a regularizer for the discriminative task, rather than to obtain an exact posterior. Therefore, even the approximate dominated by an accurate mean, it can still offer a smoother and richer supervisory signal than relying solely on a one-hot label.

## A.3 DATASET

- **CUB-200** Wah et al. (2011) is a widely-used fine-grained dataset for bird species classification. It comprises 200 bird species with 5,994/5,749 samples for training/testing.
- **ObjectNet** Barbu et al. (2019) is a challenging dataset designed to evaluate object recognition robustness in real-world scenarios. It contains 50,000 images of 313 classes for out-of-the-distribution evaluation.
- **ADE20K** Zhou et al. (2017) is a densely annotated scene parsing dataset for the semantic segmentation task. It contains 20,210/2,000 images divided into 150 object and stuff categories for training/validation.
- **MS COCO** Lin et al. (2014) is a large-scale dataset contains 80 object categories with pixel-wise and bounding box annotations. It contains 118,287 and 5,000 images used for training and validation. For open-vocabulary detection, following Gu et al. (2022), the object categories is split into 48 base and 17 novel.
- **LViS v1.0** Gupta et al. (2019) is a long-tail distribution benchmark containing 100,000/19,800/20,000 for `train`/`val`/`test`. Following prior work for open-vocabulary object detection Gu et al. (2022), the model is trained on 461 common and 405 frequent classes. The rest 337 rare classes are considered as novel and used for testing.
- **NYUv2** Silberman et al. (2012) is a popular benchmark for indoor scene understanding. It contains RGB-D images captured using a Microsoft Kinect sensor in 464 indoor environments. Following existing work, 24,231/652 image-depth pairs are used for training/validation.
- **ImageNet** Deng et al. (2009) is a large-scale dataset commonly used for object recognition. It contains 1.2M images for training and 50,000 for validation, covering a wide range of 1,000 categories.
- **CelebA-HQ** Karras et al. (2017) is a high-quality version of the CelebA dataset, comprising 30,000 images at a resolution of 1024×1024 pixels. It is widely used in computer vision research areas like image generation, super-resolution, and face synthesis.
- **LSUN-Church** Yu et al. (2015) is a subset of the Large-scale Scene Understanding (LSUN) dataset which focuses specifically on outdoor church scenes. It contains over 126,000 high-resolution images, each resized such that the shorter side measures 256 pixels.

### A.4 IMPLEMENTATION DETAILS FOR TASK-SPECIFIC DECODERS

The task-specific decoders are designed following representative work. Specifically, the classification head for visual recognition is a single-layer MLP. To enable generalization to out-of-the-distribution classes, the model computes the similarity between pooled features and text embeddings of class labels. For semantic segmentation, GENREP leverages Mask2FormerCheng et al. (2022), and calculates cosine similarities between class queries and label embeddings for open-vocabulary prediction. In open-vocabulary object detection, we follow Wu et al. (2023a) to adopt a region proposal network, and map region features into pseudo words, which are then compared with class labels. The design of the object decoder follows Zhang et al. (2024) which utilizes a DETR-style Transformer-decoder with 6 layers each containing 8 attention heads and a hidden dimension of 256. For depth estimation, we follow Li et al. (2024c) which employs a MaskFormer-like architecture, and predicts the depth value as a linear combination of bin centers.

### A.5 MONTE CARLO APPROXIMATION

We study the impact of varying sampling interval $k$ while keeping the total number of samples used for approximation constant in Table 16. As shown, when the number of samples is held constant, performance consistently improves with a larger stride $k$. This validates that more independent samples (larger $k$) yield a better distributional approximation. It also confirms that the performance decline in Table 9 is caused by the diminishing sample size, not inherent flaw in the thinning strategy.

Table 16: Analysis of the thinning interval $k$ with fixed number of sampled intermediate states.

| $k$ | $N_{\text{sample}}$ | $T$ | ObjectNet (Top-1 Acc ↑) | ADE20K (mIoU ↑) |
|---|---|---|---|---|
| 2 | 75 | 75*2+50=200 | 51.1 | 32.5 |
| 3 | 75 | 75*3+50=275 | 51.6 | 33.0 |
| 4 | 75 | 75*4+50=350 | **51.9** | **33.2** |

Since intermediate outputs of reverse diffusion are noisy or partially denoised versions of the data, it may cause mismatch to the target distribution $p(\boldsymbol{x}|y)$. We explore two strategies to mitigate this: **i)** discarding the first $m$ samples that noised heavily (*i.e.*, burn-in); **ii)** importance re-weighting to assign higher weights to later denoising steps in Equation 2. For importance re-weighting, we explore 3 approaches, which are Linear Scaling (LS), Exponential Scaling (ES), and Power Scaling (PS):

$$\text{LS: } w_t = \frac{t}{\sum_{i=1}^{T} i} = \frac{t}{\frac{T(T+1)}{2}}, \quad \text{ES: } w_t = \frac{e^{(t-1)}}{\sum_{i=1}^{T} e^{(i-1)}}, \quad \text{PS: } w_t = \frac{t^p}{\sum_{i=1}^{T} i^p}. \quad (17)$$

The experimental results are summarized below, with the thinning interval $k = 2$, power factor $p = 2$. As observed, importance re-weighting leads to poor performance, possibly due to over emphasis on a small number of samples.

Table 17: Analysis of different important re-weight approaches for sample aggregation.

| re-weighting | **N/A** | LS | PS | ES |
|---|---|---|---|---|
| ObjectNet | **51.1** | 49.2 | 49.8 | 48.7 |
| ADE20K | **32.5** | 29.3 | 30.0 | 29.1 |

### A.6 ABLATION ON HYPERPARAMETER

The key hyperparameters of GENREP are the task weights ($w_p$, $w_g$ in Eq. 17) and the alignment damping factor ($\alpha$ in Eq. 14-15). We ablate these hyperparameters below. As shown, the performance is relatively robust to minor variations. To obtain a balanced performance between perception and generation, we set $w_p = 0.7$, $w_g = 0.3$, and use the squared formulation for $\alpha$.

### A.7 PSEUDO CODE

For easier understanding, we provide the pseudo code for generative visual perception learning with knowledge distillation in Algorithm 1.

Table 18: Analysis of task weights ($w_p$, $w_g$ in Eq. 16) and the damping factor ($\alpha$ in Eq. 13-14).

| $w_p$ | $w_g$ | $\alpha$ | ObjectNet (Top-1 Acc↑) | ImageNet 256 (FID↓) |
|-------|-------|----------|------------------------|---------------------|
| 0.7 | 0.3 | $(*)^2$ | 51.1 | 6.92 |
| 0.6 | 0.4 | $(*)^2$ | 50.8 | 6.84 |
| 0.8 | 0.2 | $(*)^2$ | 51.5 | 7.12 |
| 0.7 | 0.3 | $(*)^1$ | 50.7 | 6.98 |
| 0.7 | 0.3 | $(*)^3$ | 51.3 | 7.04 |

---

**Algorithm 1** Generative Visual Perception Learning via Knowledge Distillation.

---

1: **Hyperparameters:**
2: $T \leftarrow$ total diffusion steps
3: $k \leftarrow 2$ {Thinning interval}
4: $m \leftarrow 50$ {Burn-in steps}
5: **Initialize models:**
6: diffusion_model $\leftarrow$ PretrainedDiffusionModel()
7: task_decoder $\leftarrow$ TaskSpecificDecoder()
8: hot_params $\leftarrow$ diffusion_model.attention_blocks[:]
9: **Freeze all parameters except attention blocks:**
10: freeze_all_parameters(diffusion_model)
11: unfreeze_parameters(hot_params)
12: **for** each $(x, y_{\text{true}})$ in training_data **do**
13:     **Step 1: Reverse diffusion process**
14:     $x_T \leftarrow$ sample_noise($x$)
15:     reverse_samples $\leftarrow \emptyset$
16:     **for** $t = T, T-1, \ldots, 1$ **do**
17:       $x_t \leftarrow$ diffusion_model.reverse_step($x_t, t, y_{\text{true}}$)
18:       **if** $t < T - m$ **and** $T \mod k = 0$ **then**
19:         reverse_samples.append($x_t$)
20:       **end if**
21:     **end for**
22:     **Step 2: Estimate** $p(x|y)$
23:     $\mu_{\text{list}} \leftarrow \{(s.\text{mean}) \mid s \in$ reverse_samples$\}$
24:     $\sigma_{\text{list}} \leftarrow \{(s.\text{variance}) \mid s \in$ reverse_samples$\}$
25:     $p(x|y) \leftarrow 0$
26:     **for** $\mu, \sigma \in (\mu_{\text{list}}, \sigma_{\text{list}})$ **do**
27:       $p(x|y) \leftarrow p(x|y) + \mathcal{N}(\mu, \sigma)$     (Add Gaussian component)
28:     **end for**
29:     $p(x|y) \leftarrow p(x|y)/(T//k)$
30:     **Step 3: Compute generative posterior** $p(y|x)$
31:     prior $\leftarrow 1/$num_classes
32:     logits$_{\text{gen}} \leftarrow p(x \mid y).\text{log\_prob}(x) + \log(\text{prior})$
33:     $p(y \mid x) \leftarrow$ softmax(logits$_{\text{gen}}$)
34:     **Step 4: Compute discriminative probability** $q(y|x)$
35:     logits$_{\text{disc}} \leftarrow$ task_decoder($x$)
36:     $q(y \mid x) \leftarrow$ softmax(logits$_{\text{disc}}$)
37:     **Step 5: Loss computation**
38:     loss$_{\text{disc}} \leftarrow$ cross_entropy($q(y \mid x), y_{\text{true}}$)
39:     loss$_{\text{gen\_distil}} \leftarrow$ KL_divergence($p(y \mid x), q(y \mid x)$)
40:     total_loss $\leftarrow$ loss$_{\text{disc}}$ + loss$_{\text{gen\_distil}}$
41:     **Step 6: Backpropagation**
42:     optimizer.zero_grad()
43:     total_loss.backward()
44:     optimizer.step(hot_params, task_decoder)
45: **end for**

---

