# OpenReview forum: "Reconciling Visual Perception and Generation in Diffusion Models"
_ICLR.cc/2026/Conference — ICLR 2026 Poster_

### Official Review · Reviewer_poem · 2025-10-28

**Soundness:** 3
**Presentation:** 3
**Contribution:** 3
**Rating:** 6
**Confidence:** 5

**Summary:**

The paper proposes a unified framework, **GENREP**, that jointly trains discriminative visual understanding and generative image synthesis in a single run:
(1) It uses a *Monte Carlo approximation along reverse diffusion trajectories* to distill distributional knowledge from a diffusion model, treating it as Bayesian-posterior evidence to aid downstream perception;
(2) On the generative side, it introduces *semantics-driven generation*, where semantic embeddings from the perception branch jointly modulate both the mean and the variance to steer the reverse process toward target semantics;
(3) It applies *symmetric gradient alignment* between perception and generation losses to mitigate multi-objective conflicts, forming a mutually reinforcing loop.
The authors report strong results on classification, depth estimation, open-vocabulary segmentation/detection, and class-conditional generation.

**Strengths:**

1. The motivation to unify and mutually promote visual perception and visual generation tasks by leveraging their different modeling characteristics is excellent.
2. In terms of implementation, the generative capability is effectively used to construct a sample distribution to guide perception, while the feature extraction capability from the understanding branch is used to assist denoising-based generation. This approach makes good use of the valuable information from both tasks.
3. The experiments are comprehensive. The paper provides detailed comparisons for both perception and generation, along with multi-level ablation studies that successfully demonstrate the effectiveness of the proposed solution.

**Weaknesses:**

1. Using the generative prior `p(x|y)` to estimate the posterior `p(y|x)` is inaccurate. The diffusion loss primarily learns the distribution of `x` given a specific condition `y`, and the vast majority of the learning signal is dominated by the mean, not the variance. (As demonstrated in "Improved Denoising Diffusion Probabilistic Models," the supervision signal for the variance must be small, otherwise, the generation quality degrades). This strongly suggests that the variance modeling is inaccurate. More intuitively, the mean is far more important than the variance in diffusion-based generative modeling. Furthermore, the assumption of a uniform distribution for `y` itself is questionable. The pseudocode in the appendix indicates that during training, `p(y'|x)` for all classes `y'` is treated as equally probable, rather than being reestimated from `p(x|y')`. This is clearly inaccurate in cases where there are multiple most-likely labels `y`, and thus it cannot serve as an effective distributional supervision signal.
2. The perception task is trained only on clean samples. It is uncertain whether the model can maintain good representation capabilities when applied to noised latents.

**Questions:**

1. How does the generative distillation perform on samples where there are multiple (or ambiguous) most-likely labels `y`? Is the estimated distribution accurate in such cases?

---

> ### Author Response · Authors · 2025-11-20
>
> **Q1.1: Using the generative prior $p(x|y)$ to estimate the posterior $p(y|x)$ is inaccurate.**
>
> **A1.1:** We sincerely thank you for this insightful comment. Currently, the $p(y|x)$ derived from the estimated $p(x|y)$ acts as a regularizer for the discriminative task, rather than to obtain an exact posterior.
>
> Standard discriminative losses often encourage overly confident, sharp decision boundaries, which may hinder generalization (L36–39). In contrast, our generative distillation loss guides the model output toward a distribution $p(y|x)$ that reflects the data manifold learned by the diffusion model. Therefore, even the approximate dominated by an accurate mean, it can still offer a smoother and richer supervisory signal than relying solely on a one-hot label.
>
> Your comment rightly highlights the limitations of relying on an approximate posterior. We truly appreciate this point, and have revised the limitation discussion.
>
>
> **Q1.2: The assumption of a uniform distribution for $y$ itself is questionable.**
>
> **A1.2:** Thank you for the thoughtful comment. The adoption of a uniform class prior ($(p(y) = 1/|Y|)$) is a standard practice in many machine learning problems [1,2,3].
> The core motivation is to create a non-informative prior to ensure the posterior $p(y|x)$ is shaped almost entirely by the data-driven likelihood $p(x|y)$.
>
> In large-scale modern datasets, the influence of the class prior is diminished by the vast amount of data. In such cases, a uniform prior remains neutral and avoids introducing biases. Conversely, imposing a non-uniform prior (e.g., based on sample frequencies) could overweight majority classes and harm generalization, especially for rare classes.
>
> A uniform prior is also beneficial when multiple labels are plausible.
> Under Bayes' theorem (Eq. 3), if an image $x$ is similarly consistent with 'wolf' or 'husky', the diffusion model would assign high likelihoods to both $p(x|y=\text{wolf})$ and $p(x|y=\text{husky})$.
> With a uniform $p(y)$, the posterior distributes probability across these likely classes. This creates a **soft supervision** signal rather than forcing a **single hard label**, an advantage over one-hot encoding.
> A non-uniform prior would distort this relationship. For example, inflating the posterior for 'husky' merely because it appears more frequently, even when the image favors 'wolf'.
>
> We have added additional context to clarify the role of the uniformly distributed $p(y)$ in the manuscript.
>
> [1] Semi-supervised learning with deep generative models, NeurIPS14.
>
> [2] Bayesian generative active deep learning, ICML19.
>
> [3] Gmmseg: Gaussian mixture-based generative semantic segmentation models, NeurIPS22.
>
>
>
> **Q2: It is uncertain whether the model can maintain good representation capabilities when applied to noised latents.**
>
> **A2:** We sincerely thank the reviewer for this insightful question. To empirically assess the robustness to noised latent, we evaluated GENREP on ObjectNet by feeding the perception module latents corrupted with t=10, t=20, and t=50 diffusion steps, rather than clean latents (t=0). A strong non-diffusion discriminative baseline (i.e., Swin-Transformer) is included in comparison. The results are summarized below.
>
> | Model | t=0 | t=10| t=20 | t=50 |
> | :--- | :---: | :---: | :---: | :---: |
> | Swin-Transformer (Baseline) | 40.3 | 23.1 | 11.5 | 4.6 |
> | **GENREP (Ours)** | **51.1** | **48.7** | **44.5** | **37.2** |
>
> These results provide strong empirical evidence for the robustness of the representations learned by GENREP. The robustness stems directly from our model design. The perception module uses the denoising U-Net backbone, which is inherently trained to extract semantic structure from noisy inputs. Consequently, it is well-suited to operate on corrupted latents.
> Furthermore, the conditional distribution $p(x|y)$ for knowledge distillation aggregates noised states throughout the reverse diffusion. This encourages the model to learn features from noisy latents that are predictive of the correct semantic labels.
>
> This new table and its corresponding analysis have been added to $\S 4.3$.

---

> ### Author Response · Authors · 2025-11-20
>
> **Q3: How does the generative distillation perform on samples where there are multiple (or ambiguous) most-likely labels y? Is the estimated distribution accurate in such cases?**
>
> **A3:** This scenario highlights one of the core strengths of our generative distillation compared to traditional discriminative methods.
> Standard discriminative classifiers trained with cross-entropy are pushed to output a sharp probability distribution peaking at a single ground-truth label. This overlooks the semantic similarity between concepts (e.g., "wolf" and "husky" are similar in appearance), and leads to overconfident predictions.
>
> Our approach instead models the class-conditional likelihood $p(x|y)$. If an input fits multiple classes, the diffusion model assigns high likelihoods to each of them (e.g., both wolf and husky).
> As such, applying Bayes’ rule (Eq. 3) naturally yields a **softer and more faithful posterior that reflects this ambiguity**, rather than **collapsing it into a single overconfident prediction**.
>
> Our analysis of Expected Calibration Error (ECE) provides empirical evidence for this capability (Table 10), where the incorporation of our generative distillation loss leads to a large reduction in ECE across all four tested datasets. This demonstrates the improved calibration and accuracy of the inferred posterior distribution in ambiguous setups. The explanation of Eq. ${\color{red}5}$ (i.e., $\mathcal{L}_\text{gen-distil}$) has been enriched with the discussion above.
>
> ---
>
> We sincerely appreciate your valuable time and constructive feedback. If you have any further questions, please feel free to reach out, and we will do our best to provide clarification.

---

### Official Review · Reviewer_aqJQ · 2025-11-01

**Soundness:** 3
**Presentation:** 4
**Contribution:** 3
**Rating:** 6
**Confidence:** 4

**Summary:**

This paper proposes a unified model using GenRep pipeline that jointly learns discriminative and generative modeling with a single diffusion model. The paper aims for mutual enhancement and achieves this by jointly optimizing the two objectives. For example, for perception, the model distills distributional knowledge from the generative diffusion process using Monte Carlo approximation to better approximate conditional distributions. For generation, learned high-level representations are used to guide the reverse diffusion process. Then gradient alignment mechanism is used to resolve conflicts in the updates.

**Strengths:**

- High-performing framework: GENREP successfully combines discriminative and generative paradigms without sacrificing either performance, achieving state-of-the-art or highly competitive results on both perception and generation, across various datasets
- Mutual learning synergy: The dual-branch setup intuitively builds a feedback loop — discriminative features improve the quality and faithfulness of generated images, while generative modeling provides richer distributional supervision for perception tasks. This is supported by the improved performance on downstream tasks
- Generalization across domains: The model exhibits strong out-of-distribution performance (e.g., ObjectNet) and robustness across task types, indicating that the shared latent space supports better transferability and adaptability.
- Clean optimization design: The proposed gradient alignment mechanism provides a clear way to balance conflicting gradients, promoting stable joint optimization.

**Weaknesses:**

- Computational overhead: While GENREP achieves impressive performance, integrating a diffusion model for perception tasks increases computational cost and reduces inference efficiency (e.g., 12.6 FPS vs. MaskFormer’s 19.6 FPS). This may limit its applicability in latency-sensitive or large-scale deployment scenarios.
- Approximation sensitivity and bias–variance trade-off: The performance of the generative visual perception component depends on the strided sampling interval (k) used in the Monte Carlo approximation. While a small k (e.g., 1) leads to correlated samples, increasing k beyond 2 reduces correlation but introduces higher variance and less accurate distributional estimates. The authors observe that performance peaks at k = 2 and degrades for larger k values, which is unintuitive because larger strides should, in theory, produce more independent samples and better approximate the true distribution.
- Multimodal Scalability: GENREP focuses on purely visual integration, but extending this framework to text-conditioned or cross-modal setups could further test its generality. The model currently does not explore alignment with language-conditioned diffusion models.

**Questions:**

- Training stability and sensitivity: While the paper reports stable convergence, the model jointly optimizes several interdependent losses (perception, generation, distillation, and alignment). How sensitive is GENREP to hyperparameter choices and architectural balance between these objectives? Did the authors observe instability or mode collapse when tuning these components across different datasets or backbones?
- Sampling interval (k) trade-off: Table 9 suggests performance peaks at k = 2 but declines when k increases further. Could the authors elaborate on the underlying cause of this bias–variance trade-off? Intuitively, higher k should yield more independent samples — is the degradation due to sparse coverage of the diffusion trajectory or reduced effective sample size?
- Could the GENREP architecture be extended to incorporate text-conditioned or open-vocabulary supervision (e.g., via CLIP embeddings)? How might the gradient alignment mechanism adapt in that scenario?

---

> ### Author Response · Authors · 2025-11-20
>
> **Q1: GENREP increases computational cost which may limit applicability in latency-sensitive or large-scale deployment scenarios.**
>
> **A1:** Thank you for your insightful comment. We agree that the computational overhead is an important consideration for practical applications.
>
> Our primary goal with GENREP is to introduce a unified framework that supports both visual perception and generation, while demonstrating the mutual benefits between these tasks.
> The superior performance on perception (+2.5% mIoU in Table 4 for semantic segmentation, +4.4% AP$^n_{50}$ in Table 5 for object detection) and generation (-0.85 FID in Table 6 for image synthesis) tasks compared to prior SOTA justifies the exploration of this research direction, though the inference speed is slower than conventional perception-only models.
> We discuss this performance-efficiency trade-off in $\S$A.2, which reveals a broader trend in leveraging large-scale pre-trained models, such as latent diffusion models, which, despite their higher computational demands, exhibit unmatched advantages in generalization and fine-grained understanding.
>
> To address your concern and provide a more balanced perspective, we have expanded the discussion in $\S$A.2 to more explicitly frame this issue as a trade-off and highlight efficiency optimization as a promising direction for future research.
> We hope this revision offers a more comprehensive discussion of both the current capabilities and future potential of our approach.
>
>
>
> **Q2: Performance peaks at $k = 2$ and degrades for larger $k$ values, which is unintuitive.**
>
> **A2:** Thank you for this insightful question.
> Our initial hypothesis (L461-462) is that, while a larger $k$ breaks the sample correlation, it also reduces the number of samples available for the estimate within a fixed number of diffusion steps, leading to high variance.
>
> To rigorously evaluate this hypothesis and decouple these two factors, we conduct a new experiment with varying sampling interval $k$ while keeping the total number of samples used for approximation constant.
> We achieve this by proportionally increasing the total reverse diffusion steps $T = k * N_\text{samples}+m$, where $m$ is the burn-in steps, which discards the first $m$ uninformative samples of the reverse chains (Table 14). The results are given below.
>
> | Stride ($k$) | Total Steps ($T$) | Samples ($N_\text{samples}$) | ObjectNet (Top-1 Acc) | ADE20K (mIoU) |
> | :---: | :---: | :---: | :---: | :---: |
> | 2 | 75*2+50=200 | 75 | 51.1% | 32.5% |
> | 3 | 75*3+50=275 | 75 | 51.6% | 33.0% |
> | 4 | 75*4+50=350 | 75 | **51.9%** | **33.2%** |
>
> As shown, when the number of samples is held constant, performance consistently improves with a larger stride $k$. This finding validates that more independent samples (from larger $k$) yield a better distributional approximation. It also confirms that the performance decline in our original experiment (Table 9) is caused by the diminishing sample size, not an inherent flaw in the striding strategy.
>
> We have integrated this new analysis into the Appendix and revised Section 4.3 to clarify this crucial point.
>
>
>
> **Q3: Extending this framework to text-conditioned or cross-modal setups could test its generality.**
>
> **A3: ** Thank you for this forward-looking suggestion. Extending our framework to diverse multimodal setups is indeed a compelling direction to demonstrate its generality.
> While our experiments deliberately focus on the visual modality to validate the core contributions in a clear and controlled setting, the GENREP framework is well-suited for multimodal extension.
>
> For perception tasks like classification and open-vocabulary detection/segmentation, the current conditional input $y$ is the text embedding of class labels. This mechanism can be seamlessly extended to accommodate more complex inputs, such as descriptive phrases for language-referring segmentation or even audio features for audio-visual tasks, without incurring architectural changes.
> On the other hand, the image generation branch can readily incorporate various conditioning modalities by adopting the standard paradigms of its underlying diffusion model backbone. For example, it can be conditioned on text sentences, audio clips, or other signals by integrating their embeddings through the native cross-attention mechanism.
>
> Moreover, our proposed gradient alignment strategy is modality-agnostic. It is designed to reconcile the optimization objectives of any perception and generation task, making it suited to manage the joint learning of modality-grounded perception and modality-conditioned generation.
>
> The conclusion section has been revised to better highlight the existing multimodal capabilities of our framework and explicitly position the extension to multimodal setups as a promising direction for future work.

---

> ### Author Response · Authors · 2025-11-20
>
> **Q4: How sensitive is GENREP to hyperparameter choices and architectural balance between objectives? Are there instabilities or mode collapse when tuning GENREP across different datasets or backbones?**
>
> **A4:**  The key hyperparameters of GENREP are the task weights ($w_p$, $w_g$ in Eq. 17) and the alignment damping factor ($\alpha$ used in Eq. 14-15). We conduct an ablative study on these hyperparameters below.
>
> | $w_p$ |$w_g$ | $\alpha$ | ObjectNet (Top-1 Acc) | ImageNet 256 (FID) |
> | :---: | :---: | :---: | :---: | :---: |
> | 0.7 | 0.3 | (*)$^2$ | 51.1% | 6.92 |
> | 0.6 | 0.4 | (*)$^2$ | 50.8|  6.84|
> | 0.8 | 0.2 | (*)$^2$ | 51.5% | 7.12 |
> | 0.7 | 0.3 | (*)$^1$ | 50.7 | 6.98 |
> | 0.7 | 0.3 | (*)$^3$ | 51.3% | 7.04 |
>
> As shown, the performance is relatively robust to minor variations. To obtain a balanced performance between perception and generation, we set $w_p = 0.7$, $w_g = 0.3$, and use the squared formulation for $\alpha$.
>
> Regarding potential instability or mode collapse, we did not observe such issues in any of our experiments across six benchmarks for both CNN-based and transformer-based backbones.
>
> The ablation on hyperparameters has been added to the Appendix.
>
> **Q5: Could GENREP be extended to incorporate text-conditioned or open-vocabulary supervision? How might the gradient alignment mechanism adapt in that scenario?**
>
> **A5:** Thank you for this thoughtful question. We would like to clarify that GENREP is not only extendable to open-vocabulary setups but has already been designed and evaluated for them. As reported in our manuscript, GENREP achieves SOTA performance on open-vocabulary semantic segmentation (Table 4) and open-set object detection (Table 5).
>
> For these tasks, instead of a fixed classifier, we compute the cosine similarity between the visual features (e.g., pixel- or region-level embeddings) and the text embeddings of category names (L862-866), derived from the text encoder used for textual conditioning. It can also be easily adapted to incorporate CLIP features, as demonstrated by the last row of Table 4, where GENREP uses an LDM pretrained on LAION-5B with the CLIP text encoder. This illustrates its inherent compatibility with open-vocabulary supervision.
>
> Regarding gradient alignment, it requires no adaptation. The mechanism is principled and loss-agnostic. It operates on the final gradient vectors, and decomposes them into parallel and orthogonal components (Eqs. 11–12). This decomposition depends only on the gradients themselves, not on the specific loss formulation. Thus, whether the perception loss uses standard cross-entropy over fixed classes or an open-vocabulary objective based on CLIP embeddings, the method continues to mitigate conflicts and promote reconciliation in the optimization process.
>
>
> ---
>
> Thank you for your valuable time and constructive feedback. We have revised the manuscript with more discussions and experimental results according to your comments. It is welcome if there are any further questions or suggestions.

---

### Official Review · Reviewer_fxcY · 2025-11-01

**Soundness:** 3
**Presentation:** 2
**Contribution:** 2
**Rating:** 4
**Confidence:** 4

**Summary:**

This paper presents GENREP, a framework aiming to unify image understanding (perception) and image synthesis (generation). The method comprises three main components:
1) Generative Visual Perception Learning: Using Monte Carlo approximation (a mature approach from Generative Classifier) on intermediate (noisy) samples from the reverse diffusion process to estimate the class posterior $p(y|x)$, which serves as an auxiliary distillation loss $\mathcal{L}_{gen \underline{ } distil}$ to guide the perception task.
2) Semantic-Driven Generation: Using high-level semantic representations ($x_{sem}$) from the perception task to dynamically modulate the mean ($\mu_\theta$) and variance ($\sigma_\theta$) of each step in the generative process.
3) Gradient Alignment: Using a gradient projection strategy from multi-task learning to reconcile conflicts between the perception and generation objectives.

The authors claim this unified model achieves SOTA performance on numerous perception and generation benchmarks.

**Strengths:**

1. Meaningful Goal: The attempt to unify discriminative learning (perception) and generative modeling (generation) within a single framework that allows for mutual benefits is a valuable research direction.
2. Interesting Core Idea: The paper links generation and perception of a same Diffusion model via Bayes' theorem, which is a reasonable and natural synergy.
3. Extensive Empirical Evaluation: The paper validates its approach across a wide range of benchmarks, including classification, segmentation, etc, achieving competitive results.

**Weaknesses:**

1.  The Necessity of Framework: The paper claims its novelty lies in adding "high-level semantics" to overcome the "low-level reconstruction" limitations of other unified models. However, the proposed mechanisms (MC estimation of $p(y|x)$ and explicit $\mu_\theta, \sigma_\theta$ modulation) are exceptionally complex (especially for computation), and I am not aware of how the so-called "high-level semantics" are contributing or whether they are necessary. In contrast, other unified frameworks (e.g., Diff-2-in-1 [1]) appear to achieve a similar unified goal via a much simpler self-training/mean-teacher loop, even without explicit "high-level semantics". This severely questions the necessity of GENREP's convoluted approach (too much cost for limited improvement, or meaningless innovation). Also, please conduct a more comprehensive literature review and a more detailed comparison of similar unified diffusion models, like [1].
2.  Theoretically Flawed MC Approximation: The "Generative Visual Perception Learning" relies on an MC approximation of $p(x|y)$. This estimation is theoretically flawed in several major ways:
    * Correlated Samples: It uses samples from a Markov chain (the reverse diffusion process). The claim that strided sampling ($k=2$) is sufficient to "break" this correlation is unsubstantiated and highly unlikely.
    * Biased Samples: It uses *intermediate, partially-denoised* samples to approximate the *final, clean* data distribution $p(x|y)$. This is a fundamentally biased estimator.
    * Computational Overhead: As shown in Algorithm 1, this approximation must be computed *inside every training iteration*, adding massive, unanalyzed computational overhead. Can this approximation be further accelerated?


[1] Zheng, S., Bao, Z., Zhao, R., Hebert, M., & Wang, Y. X. Diff-2-in-1: Bridging Generation and Dense Perception with Diffusion Models. In The Thirteenth International Conference on Learning Representations.

**Questions:**

1.  Validity of MC Approximation: Can you provide stronger evidence that strided sampling ($k=2$) is sufficient to yield (near) independent samples from the reverse diffusion process? Furthermore, can you quantify the bias introduced by using noisy, intermediate samples to estimate the final, clean distribution $p(x|y)$?
2.  Question on "Positive Feedback Loop": I am curious to see from Table 8 (rows #1, #3), it shows that  adding Semantic-Driven Generation (Percept -> Gen) *hurts* perception (Top-1/mIoU drops). Some explanations or investigations? This is helpful for the claims of "positive feedback loop".
3.  Quantification of Training Cost: Compared to a standard perception baseline or a simpler unified framework (e.g., Diff-2-in-1), please quantify the extra training cost (e.g., in GPU-hours) introduced by the iterative Monte Carlo sampling required by your generative perception loss (Algorithm 1 in Appendix A.7)?

---

> ### Author Response · Authors · 2025-11-20
>
> **Q1.1: MC estimation and explicit $\mu_\theta$, $\sigma_\theta$ modulation are exceptionally complex (especially for computation).**
>
> **A1.1:** Thank you for your feedback. We would like to clarify that GENREP is designed as a **truly unified model** capable of mastering both understanding and generation tasks. In contrast, prior diffusion-based perception approaches optimize the models solely for visual perception. Even Diff-2-in-1 is primarily trained as a task-specific data augmenter, raising concerns about the preservation of its general-purpose image generation capabilities.
>
> **Given this unified context, it is inappropriate to compare the training cost of GENREP against perception-oriented baselines.**
> For GENREP, it can be considered as simultaneously equipping the model with visual perception capabilities during image generation learning. Compared to standard image generation training approaches, its additional cost is two sets:
>
> - First, we approximate $p(x|y)$ by reusing intermediate states from a single reverse diffusion trajectory (L181-184). This avoids the heavy cost of generating hundreds of fully denoised samples, which standard MC methods require. Therefore, our method introduces **nearly zero additional computation cost** compared to prior image generation methods.
>
> - Second, the semantic correction function ($f_\text{sem}$) is implemented as a simple learned weight matrix ($W_p$) for the mean, and a small Multi-Layer Perceptron (MLP) for the variance. These are minimal components in modern neural networks with negligible additional FLOPS.
>
> Therefore, let alone the perception head, the computation cost of GENREP is similar to merely training an image generation model. We believe the gains in both perception and generation tasks justify our design, and its computational cost remains practical. The runtime analysis in Appendix has been revised and moved to $\S 4.3$ to make these points clearer.
>
>
> **Q1.2: How high-level semantics are contributing or whether they are necessary.**
>
> **A1.2:** In this work, "high-level semantics" are concretely embodied as the rich, discriminative feature representations learned by the perception branch (L45, L68-69). Using it to instruct generative modeling holds reasonable motivation from the observation in recent work [1,2,3] that "high-quality discriminative representations accelerate the generative learning of diffusion models" (L49-50). This principle has been explored and validated in several contemporary works, forming a foundation for our approach.
>
> In our method, the perception-optimized features $x_\text{sem}$ are used to directly inform the reverse diffusion process to incorporate these high-quality discriminative features, which form the basis to create a feedback loop to enhance both generation and perception. Its necessity is verified in Table 8, where the FID score for image generation enjoys a dramatic improvement compared to a model without it (row #3 $\textit{v.s.}$ row #1).
>
> [1] Representation alignment for generation: Training diffusion transformers is easier than you think, ICLR25.
>
> [2]. Reconstruction vs. generation: Taming optimization dilemma in latent diffusion models, CVPR25.
>
> [3]. Learning diffusion models with flexible representation guidance, NeurIPS25.
>
> **Q1.3: Other unified frameworks (e.g., Diff-2-in-1) appear to achieve a similar unified goal via a much simpler self-training/mean-teacher loop.**
>
> **A1.3:** Thank you for highlighting this important related work. While work like Diff-2-in-1 also pursues a unified model, we wish to clarify that GENREP is founded on a different philosophy:
>
> - The core mechanism in Diff-2-in-1 is to generate new data samples for discriminative learning. In contrast, GENREP distills the underlying **distributional knowledge** embedded in generative models. This directly aligns discriminative learning with the data distributions captured by generative modeling, and goes beyond instance-level augmentation.
>
> - Diff-2-in-1 improves the quality of the generated data to benefit the discriminative tasks. The knowledge flow is unidirectional from generation to perception. GENREP, in contrast, establishes a **bidirectional feedback loop** for mutual enhancement of both tasks within a single model.
>
> - The mean-teacher framework uses EMA to update the teacher model. This promotes training stability but does not address potential optimization conflicts between generation and perception objectives. Our gradient alignment strategy is specifically designed to tackle this challenge, to achieve a **balanced and joint optimization** where both capabilities are enhanced without compromising the other.
>
> In summary, our goal is not simply unifying these tasks, but **reconciling** them. This fosters a balanced relationship where both capabilities can mutually benefit and co-develop. $\S 2$ has been revised to provide a more comprehensive review of unified frameworks and better contextualize our contributions.

---

> ### Author Response · Authors · 2025-11-20
>
> **Q2.1: The reliance on an MC approximation is theoretically flawed.**
>
> **A2.1:** Thank you for your constructive feedback. Our usage of intermediate states during reverse diffusion (i.e., Markov chain transitions) to estimate distribution follows a classic variant of Monte Carlo (MC) methods known as Markov Chain Monte Carlo (MCMC) [1].
>
> In contrast to MC relying on independent random samples, MCMC generates dependent samples through a Markov chain to reach the target distribution.
> The strided sampling strategy corresponds to the $\textbf{thinning}$ operation in standard MCMC, which keeps every $k$-th sample from the Markov chain, and its reliability is therefore established. To avoid the negative impact of overly noisy samples on the estimation, we also discard the first $m$ samples that are noised heavily during reverse diffusion ($\S A.6$ of the original manuscript). This is analogous to the $\textbf{burn-in}$ phase in MCMC.
>
> To enhance the theoretical clarity, we have revised $\S$3.2 to present our approach following the pipeline of MCMC.
>
> [1] Practical markov chain monte carlo, Statistical science, 1992.
>
> **Q2.2: Using strided sampling ($k=2$) to "break" correlation is unsubstantiated and highly unlikely.**
>
> **A2.2:** Thank you for raising this insightful question. While using all samples already yields satisfactory results (i.e., 50.3 when $k=1$ v.s. 51.1 when $k=2$ in Table $\color{red}9$), strided sampling serves two objectives: i) giving a set of more "independent-like" samples; ii) improving efficiency as covering every sample in a long chain can be memory- and computation-intensive.
> We have tuned down our claim from "break correlation" to "reduce correlation and improve efficiency".
>
> **Q2.3: Using intermediate samples results in a fundamentally biased estimator.**
>
> **A2.3:** Our motivation for using noisy intermediate states instead of clean independent samples is to strike a balance between **statistical purity** of the estimator and **computational feasibility**.
>
> A truly unbiased estimate of $p(x|y)$ needs hundreds of fully denoised samples at each training step. This significantly increases the training cost, and makes the joint learning infeasible.
> We thus reuse the intermediate states from a \textbf{single} reverse diffusion pass to acquire a class-conditional signal to regularize the perception model, while adding nearly **zero additional computational cost**.
>
> The strong empirical results across six perception benchmarks (Table ${\color{red}1}$-${\color{red}5}$) and image generation tasks (Table ${\color{red}6}$-${\color{red}7}$) serve as evidences that this MCMC-based estimator is effective. The reasons are as follows:
>
> - $\textbf{First}$, recent analyses reveal that during image generation, diffusion models first establish low-frequency, global semantic structures (e.g., shape, layout) and later add high-frequency, fine-grained details (e.g., texture). More interestingly, [1] finds that earlier denoising steps (larger timestep $t$ with noised features) tend to yield more semantically-aware features, while later steps (smaller timestep $t$ with clear features)  focus more on low-level details. This implies that the intermediate samples can capture multi-scale representations via generative modeling, which serve as an effective teaching signal for discriminative learning.
>
> - $\textbf{Second}$, the usage of noisy intermediate states is not merely a hypothesis, but the foundation of a growing line of research. Numerous recent studies cited in the manuscript  [1,2,3] have shown that features from intermediate steps of diffusion models are powerful backbones for downstream perception tasks (L47-48, L110-114). These work establishes a strong preliminary for the value of the intermediate signals we leverage.
>
> [1] Emergent correspondence from image diffusion, NeurIPS23.
>
> [2] Unleashing text-to-image diffusion models for visual perception, ICCV23.
>
> [3] Diffusion model as representation learner, ICCV23.
>
> **Q2.4: This approximation computed inside every training iteration adds massive computational overhead.**
>
> **A2.4:** Thank you for raising this practical concern. As explained in **A1.1** and **A2.3**, the MCMC approximation reuses the intermediate samples, which contribute to nearly **zero additional overhead** compared to the generative baseline.
>
> We further quantify the computational cost of MCMC approximation below, where "perception only" means solely visual perception learning with a single forward pass through the diffusion model.
>
> |Model|Trainable Params (M)| Training Time (GPU Hour)|Inference Time (FPS)|ADE20K (mIoU)|
> | :---: | :---: | :---: | :---: | :---: |
> |Perception only|54|79|12.6|49.3|
> | + MCMC approx. |54|87|12.6|52.2|
>
> As seen, the sole overhead of using MCMC approximation is a modest 10.1% increase in training time, which is far from "massive" and a practical trade-off for the performance improvement.

---

> ### Author Response · Authors · 2025-11-20
>
> **Q3.1: Stronger evidence that strided sampling ($k=2$) is sufficient to yield (near) independent samples.**
>
> **A3.1:** Thank you for your constructive feedback. To address your concern, we employ the Autocorrelation Function (ACF), a standard statistical method to measure the temporal correlation within a sequence of samples. A lower ACF value indicates weaker correlation.
>
> We compute the average ACF for samples separated by different thinning intervals ($k$) throughout the reverse diffusion process, averaged over all data points in the test set of ObjectNet.
> The results are summarized below.
>
> | Thinning Interval (k) | ACF | ObjectNet (Top-1 ACC)|
> | :---: | :---: | :---: |
> |1 | 0.87| 50.3|
> |2 | 0.63| 51.1|
> |3 | 0.46| 51.3|
> |4 | 0.35| 50.5|
>
>
> It reveals that larger $k$ can reduce correlation, which leads to improvements in performance (row #1-2). However, due to the limited length of sequence, larger $k$ leads to fewer samples for approximation and inferior performance (row #2-4). Therefore, rather than breaking correlation, $k=2$ is a well-justified balance between model effectiveness and efficiency. Related statement in the manuscript has been revised.
>
>
>
> **Q3.2: Quantify the bias introduced by using noisy, intermediate samples.**
>
> **A3.2:** To quantify the bias, we conduct an empirical study comparing our MCMC-based approximation against a relatively unbiased baseline constructed from a large number of fully denoised samples.
>
> Specifically, for each image in the validation set of ObjectNet and CUB-200, we estimate the posterior probability $p(y|x)$ in two ways:
>
> - i) We generate 75 fully-denoised samples by performing 75 complete reverse diffusion runs, each with random data augmentation applied to the input. We then estimate the likelihood $p'(x|y)$.
>
> - ii) We estimate the likelihood $p(x|y)$ using our proposed MCMC method, which requires only a single reverse pass.
>
> We then compute the corresponding posterior distributions $p'(y|x)$ and $p(y|x)$ using Bayes' theorem with a uniform prior. The bias is quantified as the average Kullback-Leibler (KL) divergence between these two distributions across all validation images.
>
> | Dataset  | KL($p(y\|x)\|\|p'(y\|x)$) |
> | :---- | :----: |
> | ObjectNet | 0.123 |
> | CUB200   | 0.175 |
>
> These relatively small KL divergence values indicate that the posterior distribution obtained from intermediate states is a close approximation of the one derived from the more computationally expensive fully denoised samples. This small bias is a trade-off for computational efficiency.
>
>
>
> **Q4: Adding Semantic-Driven Generation hurts perception.**
>
> **A4:** Thank you for this sharp observation. As diffusion models are trained on a pixel-level reconstruction objective, they inherently possess a bias towards low-level details to achieve high-fidelity image synthesis.
> To counterbalance this and strengthen high-level discriminative learning, we set the task weights for perception and generative as 0.7 $\textit{v.s.}$ 0.3.
>
> After using merely semantic-driven generation, the optimization shifts further toward generation, preferring more on low-level details and degrading perception performance. To verify this, we increase the perception weight to 0.8:
>
> | Setup | ObjectNet (Top-1 ACC)| ADE20K (mIoU) | ImageNet 256 (FID) |
> | :---: | :---: | :---: |:---: |
> | Baseline (0.7 $\textit{v.s.}$ 0.3) | 45.4| 27.8| 13.27|
> | Semantic-Driven Generation (0.7 $\textit{v.s.}$ 0.3) | 44.1 | 25.6| 7.45|
> | Semantic-Driven Generation (0.8 $\textit{v.s.}$ 0.2) | 46.3 | 28.9| 7.98|
>
> As seen, perception improves with higher perception weights, while generation remains much better than baseline. However, static task weights cannot resolve the conflict occurring dynamically at the gradient level for each training batch. This leads to inferior generation performance for row#3 compared to row#2. This motivates  Gradient Alignment, which acts as a dynamic and fine-grained resolver for these conflicts, and yields the best performance in row #5 of Table $\color{red}8$.

---

> ### Author Response · Authors · 2025-11-20
>
> **Q5: Quantify the extra training cost (e.g., in GPU-hours) introduced by the iterative Monte Carlo sampling.**
>
> **A5: ** Thank you for pointing this out. Our original manuscript includes a detailed runtime analysis regarding trainable parameters (M), training time (GPU Hours), and inference speed (FPS) in Table ${\color{red}13}$.
>
> | Method | Backbone | Trainable Params (M)| Training Time (GPU Hours) | Inference (FPS) | mIoU |
> | :--- | :--- | :---: | :---: | :---: | :---: |
> | DeepLabV3+ | ResNet-101 | 63 | 83 | 14.2 | 45.5 |
> | SETR | ViT-L | 308 | 623 | 9.7 | 46.2 |
> | UperNet  | Swin-S | 81 | 104 | 15.2 | 47.7 |
> | MaskFormer | Swin-S | 63 | 53 | 19.6 | 49.8 |
> | GENREP (perception only) | LDM | 54 | 79 | 12.6 | 49.3 |
> | GENREP | LDM | 54 | 87 | 12.6 | **52.2** |
>
> It reveals that with similar levels of training params, GENREP needs fewer training GPU hours compared to conventional perception baselines (e.g., UperNet with Swin-S as the backbone). As noted in **A2.4**, we have additionally included a perception-only baseline to better isolate the cost introduced by the iterative Monte Carlo sampling. The revised table has now been incorporated into the main text under the "Runtime Analysis" subsection in $\S\color{red}{4.3}$.
>
>
> ---
>
>
> We fully understand your concerns regarding i) the computation overhead over perception-oriented baselines, and ii) the reliability and technical rigor of our MC approximation.
> For i), it is noteworthy that our work aims to create a truly unified model for both visual perception and generation. Therefore, a fair assessment of its efficiency should be framed against this ambitious goal, rather than models designed for only perception tasks. For ii), we now explicitly present our method following the principles of MCMC, and supplement $\S\color{red}3$ with additional background to improve its technical justification.
>
> We are grateful for these sharp observations. We hope our explanations and revisions have adequately addressed your concerns and welcome any further questions or suggestions.

---

### Official Review · Reviewer_8E75 · 2025-11-01

**Soundness:** 3
**Presentation:** 3
**Contribution:** 3
**Rating:** 8
**Confidence:** 3

**Summary:**

This paper present GenRep, a unified image understanding and synthesis model that jointly conducts discriminative learning and generative modeling in one training session. This paper adopts a novel gradient alignment strategy to guide the joint optimization of perception and generation.

**Strengths:**

- the idea to distill distributional knowledge embedded in diffusion models to guide the discriminative learning for visual perception tasks is novel
- the experimental results are comprehensive
- the paper is well-written
- the derivation is clear

**Weaknesses:**

- the gradient alignment may introduce extra computational overhead during training.

**Questions:**

- can the proposed methods be extended to more visual understanding tasks, such as OCR, captioning?

---

> ### Author Response · Authors · 2025-11-20
>
> **Q1: The gradient alignment may introduce extra computational overhead during training.**
>
> **A1:** Thank you for pointing this out.
> Theoretically, the additional computations introduced by our gradient alignment (Eqs. ${\color{red}11}\text{-}{\color{red}16}$) are minimal, as it consists of mere vector operations (e.g., dot products, norms, additions) which are computationally negligible compared to the backward pass through the entire network.
>
> To properly address your concern, we benchmark the training time of our model with and without the gradient alignment component. The results are as follows:
>
> | Model |Trainable Params (M) | Training Time (GPU Hour) | Inference Time (FPS) | ADE20K (mIoU)|
> | :---: | :---: | :---: | :---: | :---: |
> | Baseline | 54 |  85 | 12.6 | 50.9 |
> | + Gradient Alignment | 54 |  87 | 12.6 | 52.2|
>
> As seen, enabling gradient alignment introduces a 2.4% increase in total training time, while the inference speed remains unaffected. In exchange for this marginal cost, the model performance improves by 1.3 mIoU.
>
>
>
> **Q2: Can the proposed methods be extended to more visual understanding tasks, such as OCR, captioning?**
>
> **A2:** Thank you for this forward-looking question. The core principles of GENREP are generalizable and can be extended to more complex visual understanding tasks, including OCR and image captioning. The idea of creating a feedback loop between discriminative perception and generative synthesis tasks is modality-agnostic.
>
> Specifically, for OCR, the perception branch functions as an OCR model that receives an image and outputs a character sequence. Its objective can be a standard OCR loss, while the diffusion model is trained to generate realistic text images conditioned on a given character sequence. The distributional knowledge distilled from the generator serves as a regularizer that exposes the OCR model to diverse text appearances, and improves its robustness and generalization. On the other hand, high-level semantic features extracted by the OCR branch guide the diffusion model toward synthesizing more faithful and contextually coherent text images.
>
> For image caption, the perception task corresponds to the captioning model, which interprets the visual scene and generates a descriptive word sequence. The diffusion model continues to perform image synthesis, conditioned on captions produced by the perception branch. The joint training with image generation forces the shared visual encoder to learn richer and more robust representations, which provide a stronger foundation for the language decoder to yield more accurate and detailed captions. In turn, captions supply high-level semantic guidance for image synthesis. This creates a self-consistency loop where the model learns to generate images that align with its own descriptions.
>
> ---
>
> Thank you so much for your suggestive comments. Please feel free to post your feedback if you have any further questions.

---

### Comment · Area_Chair_WRPY · 2025-11-22
**Start discussion**

Hi all,

The authors have submitted their response to the initial reviews, and we now enter the discussion phase.

Please review the authors' response and the comments from other reviewers. Based on the rebuttal and discussion, please update your final score if appropriate.

We welcome and encourage further discussion as needed.

Thank you for your continued contributions.

Best regards,

Your AC.

---

### Author Response · Authors · 2025-12-02

We express our sincere thanks to the Area Chair for their time and effort. Below, we summarize how we address the key concerns for a quick review.

**1. Theoretical Rigor: MCMC Framework Validation (R-fxcY)**
To address concerns regarding the use of correlated samples, we have explicitly reframed our approach following the **Markov Chain Monte Carlo (MCMC)** principles in $\S3$. We clarify that our strided sampling corresponds to the **chain thinning**  operation in standrard MCMC methods, which holds reasonable motivations. Empirically, we also add **Autocorrelation Function (ACF)** analysis to prove correlation reduction and measure the KL divergence against a "gold standard" (fully denoised independent samples). The low divergence value (e.g., **0.123** on ObjectNet) confirms minimal bias in our approximation.

**2. Computational Efficiency (R-8E75, R-fxcY, R-aqJQ)**
Responding to concerns about training costs, we emphasize that GENREP is a **unified model** for both generation and understanding. Compared to standard image generation training, our method introduces **nearly zero additional cost** by reusing intermediate states, which essentially equips the model with perception capabilities for free. Even against a strict "perception-only" baseline, the overhead is marginal (**~10%** training time) but yields significant gains (**+2.5% mIoU, +4.4% AP**) and high-fidelity synthesis capabilities, justifying the trade-off.

**3. Necessity & Comparison to Unified Baselines (R-fxcY)**
We distinguish our approach from simple augmentation methods like *Diff-2-in-1* by demonstrating that GENREP establishes a **bidirectional feedback loop**. We verify that removing this semantic-driven component significantly degrades generation quality (FID worsens from **6.92 to 13.27**). Additionally, we address the concern about "unintuitive" sampling trends by controlling for sample size in new experiments. The results show that performance consistently improves with larger sampling strides, validating our thinning strategy.

**4. Theoretical Grounding of Generative Distillation (R-poem)**
We clarify the probabilistic rationale for Reviewer **poem**, explaining that the estimated posterior functions as a **distributional regularizer**. Unlike rigid one-hot targets, this "soft" supervision preserves semantic ambiguity (e.g., visual similarities between "wolf" and "husky"), and provides richer discriminative signals. We further justify the use of a **Uniform Prior** to eliminate class-frequency bias, ensuring supervision is driven by the data likelihood $p(x|y)$.

**5. Robustness & Generalization (R-8E75, R-poem, R-aqJQ)**
To demonstrate robustness, we conduct new experiments which show GENREP can significantly outperform non-diffusion baselines (e.g., Swin-T) on corrupted latents, and achieve **37.2% vs. 4.6%** accuracy at $t=50$. We also emphsize the modality-agnostic nature of GENREP, confirming it is readily extensible to multimodal tasks and adapted to open-vocabulary tasks.

---

### Meta-Review · Area_Chair_8PKe · 2025-12-31

**Summary:**

This paper proposes GENREP, a framework designed to reconcile visual perception and generation in diffusion models by establishing a mutual feedback loop between the two capabilities. Most of technical and theoretical concerns are resolved via supplementary experiments, theoretical revisions, and quantitative analyses. Question about extending the method to visual understanding tasks (e.g., OCR, image captioning) is addressed. Concerns about computational feasibility, unintuitive performance trends with increasing k, and cross-modal extensibility are addressed.

However, the reviewer aqJQ's concerns about extending GENREP to open-vocabulary supervision and the adaptation of the gradient alignment mechanism is not sufficiently resolved despite supplementary explanations. The authors need to further provide revisions in the final version.

Based on the above considerations, I think the current manuscript basically meets the requirements of ICLR and I recommend to accept this manuscript.

**Reviewer Concerns:**

Most of technical and theoretical concerns are resolved via supplementary experiments, theoretical revisions, and quantitative analyses. Reviewer 8E75's question about extending the method to visual understanding tasks (e.g., OCR, image captioning) is addressed. Reviewer fxcY's multiple concerns are effectively resolved. Reviewer aqJQ's concerns about computational feasibility, unintuitive performance trends with increasing k, and cross-modal extensibility are addressed.

**Reviewer Scores:**

Reviwer fxcY may raise the score and other reviewers would keep their positive scores.

---

### Decision · Program_Chairs · 2026-01-26

Accept (Poster)